# Neuropathological correlates and genetic architecture of microglial activation in elderly human brain

Daniel Felsky[1,2], Tina Roostaei [1], Kwangsik Nho[3], Shannon L. Risacher[3], Elizabeth M. Bradshaw[1], Vlad Petyuk [4], Julie A. Schneider[5,6], Andrew Saykin [3], David A. Bennett[5,6] & Philip L. De Jager [1,2]

Microglia, the resident immune cells of the brain, have important roles in brain health. However, little is known about the regulation and consequences of microglial activation in the aging human brain. Here we report that the proportion of morphologically activated microglia (PAM) in postmortem cortical tissue is strongly associated with β-amyloid, tau-related neuropathology, and the rate of cognitive decline. Effect sizes for PAM measures are substantial, comparable to that of *APOE* ε4, the strongest genetic risk factor for Alzheimer's disease, and mediation models support an upstream role for microglial activation in Alzheimer's disease via accumulation of tau. Further, we identify a common variant (rs2997325) influencing PAM that also affects in vivo microglial activation measured by [[11]C]-PBR28 PET in an independent cohort. Thus, our analyses begin to uncover pathways regulating resident neuroinflammation and identify overlaps of PAM's genetic architecture with those of Alzheimer's disease and several other traits.

[1] Center for Translational and Computational Neuroimmunology, Department of Neurology, Columbia University Medical Center, 630 West 168th Street, New York, NY 10032, USA. [2] Program in Population and Medical Genetics, Broad Institute of MIT and Harvard, 320 Charles Street, Cambridge, MA 02141, USA. [3] Indiana Alzheimer's Disease Center, Center for Neuroimaging, Department of Radiology and Imaging Sciences, Center for Computational Biology and Bioinformatics, Indiana University School of Medicine, 355 West 16th Street, Indianapolis, IN 46202, USA. [4] Pacific Northwest National Laboratory, Richland, WA 99354, USA. [5] Department of Neurology, Rush University Medical Center, 1653 West Congress Parkway, Chicago, IL 60612, USA. [6] Rush Alzheimer's Disease Center, Rush University Medical Center, 1653 West Congress Parkway, Chicago, IL 60612, USA. Correspondence and requests for materials should be addressed to P.L.D.J. (email: pld2115@cumc.columbia.edu)

The function of immune cells in the central nervous system (CNS) has recently become a major focus in human genetics, as these cells have been implicated in susceptibility to neurodegenerative, autoimmune, and psychiatric diseases. Microglia, the brain's resident immune cells, are thought to have important roles in both tempering and exacerbating aging-related neuropathological processes, but their precise role remains unclear as they are difficult to access in human subjects. Recently, a molecularly defined subtype of disease-associated microglia has been proposed to exist in a mouse model of Alzheimer's disease (AD)[1]. However, transcriptomic identities of isolated microglia are notoriously plastic[2] and highly susceptible to a myriad of experimental confounds[3]. Regional and temporal heterogeneity of microglia subpopulations have also been shown in human and mouse models based on both molecular and morphological characteristics.

Recent postmortem studies have shown that microglial densities in specific regions are associated with a syndromic diagnosis of both early and late-onset AD[4], and a recent systematic review of 113 studies quantifying microglial activation in postmortem AD brain highlighted the importance of activation vs. abundance of these cells in disease[5]. However, low sample sizes, indirect measures of microglia, and lack of full antemortem and postmortem pathological assessments all limit the insights that can be drawn from the individual component studies and this systematic review. Here, we leverage two large cohort studies of cognitive aging that include antemortem longitudinal cognitive assessments and structured postmortem histopathological evaluations to characterize a postmortem measure of microglial activation, directly observed by immunohistochemical staining and light microscopy. This morphological assessment of microglial activation stage represents a clear and robust measurement of neuroinflammation that cannot be captured by a surrogate marker. We first examine how this measure relates to different aging-related pathologies. We follow this with causal mediation analyses aimed at placing microglial activation temporally within the cascade of pathological events leading to AD. Finally, we perform genome-wide analyses to identify the genomic architecture of microglial activation and implement a high-resolution polygenic risk scoring method based on Mendelian randomization assumptions to demonstrate putatively causal effects of microglial activation on multiple human diseases and traits. Figure 1 illustrates the set of analyses performed in our study.

## Results

**Active microglia discriminate pathological AD.** The characteristics of ROS/MAP participants with microglial count data are presented in Table 1. We first performed pairwise Spearman correlations of each individual microglial density measurement followed by hierarchical clustering (Supplementary Figure 1), finding that stage I microglial densities were more similar between regions, whereas stage II and III microglial densities were more highly correlated within cortical and subcortical regions separately.

Following this observation and prior reports of the presence of morphologically defined active microglia in much smaller samples of AD brains ($n$ range = 34–106)[5], we performed benchmarking discriminatory analyses for each microglial density phenotype (see Fig. 2a for illustrative examples of staged microglia) in two steps. First, Welch $t$-tests comparing mean measures of microglial density between individuals with a postmortem pathological diagnosis of AD vs. non-AD found that total microglial density as measured in any region was not discriminative of AD status ($0.33 < p < 0.68$; Fig. 2b). On the other hand, stage III microglial density was different between AD and non-AD subjects, though this was only true in cortical regions (midfrontal (MF) $p = 1.5 \times 10^{-8}$, Cohen's $d_{[95\%\,CI]} = 0.80$ [0.52,1.08]; inferior temporal (IT) $p = 6.4 \times 10^{-9}$, Cohen's $d_{[95\%\,CI]} = 0.84[0.55,1.12]$). Further, a stronger association was observed with the proportion of stage III microglia density relative to total microglia (the proportion of active microglia, or PAM) (MF $p = 1.8 \times 10^{-10}$, Cohen's $d_{[95\%\,CI]} = 0.91[0.63,1.19]$; IT $p = 1.5 \times 10^{-11}$, Cohen's $d_{[95\%\,CI]} = 0.99[0.70,1.28]$), confirming that morphologically activated microglia rather than the total number of microglia is most important for the accumulation of AD-related pathology in aging[6,7]. This is consistent with earlier work in mice demonstrating that microglial activation rather than proliferation mediates neurodegeneration[8].

Second, logistic regression modeling of pathological AD in the same sample confirmed our $t$-test results, finding that models including PAM outperformed other models with a maximum AUC of 0.795 for IT and 0.792 for MF (Fig. 2c; pathological distributions between pathological AD and non-AD subjects are described in Supplementary Data 9). Notably, the effects of both cortical PAM measures were independent of and improved model performance to a greater extent than the major *APOE* ε4 genetic risk factor for AD. In the case of IT, the inclusion of PAM increased model performance over the co-variate only model by 18% (AUC = 0.745 vs. 0.565), whereas including *APOE* ε4 status only yielded a 9.6% increase (AUC = 0.661). Bootstrap analyses reinforced these findings, with calibrated PAM-inclusive models showing 21.4% (IT) and 20.4% (MF) increases in model accuracy vs. the *APOE* ε4 status-inclusive model's 11.9%. In full models containing both *APOE* ε4 status and PAM, the effect sizes of each were comparable for a difference in PAM of one interquartile range (IQR) (MF OR$_{APOE\varepsilon4}$ = 6.9[2.73,17.4], OR$_{PAMIQR[95\%CI]}$ = 4.8[2.78,8.15]; IT OR$_{APOE\varepsilon4[95\%CI]}$ = 6.5[2.39,17.64], OR$_{PAMIQR[95\%CI]}$ = 4.2[2.46,7.15]). Importantly, neither *APOE* ε4 nor ε2 status were related to either cortical PAM measure, whether or not co-variates were included (all $p > 0.1$).

**PAM effects on neuropathology and cognitive decline.** Having found robust effects of PAM on pathologically defined AD, and given the high degree of neuropathological heterogeneity found in our cohort[9,10], we sought to identify whether effects of PAM were specific to β-amyloid (Aβ) and tau (the defining pathologic characteristics of AD) or were also associated with other neuropathological features commonly observed in aged individuals. Using robust regression across 18 pathologies, we found that cortical PAM measures were associated to the greatest extent with total Aβ load and neuritic amyloid plaque count, and, to a lesser but still significant extent, with paired helical filament (PHF) tau, neurofibrillary tangles, and diffuse plaques (Fig. 3a). All relationships were in the positive direction, whereby an increase in PAM paralleled an increase in pathology. Notably, we found no significant associations of either subcortical PAM with any pathology, nor did we find association of any PAM measure with pathologies not related to Aβ or tau accumulation (Fig. 3a). Tests of person-specific linear trajectories of cognitive decline revealed significant associations of IT PAM with global cognitive decline, as well as with decline in all five cognitive sub-domains (Fig. 3b). MF PAM was also associated with global cognitive decline, and four of the five cognitive sub-domains. Similar to our neuropathological findings, there were no significant associations of subcortical PAM with cognitive decline.

To test whether associations were missed due to a lack of regional specificity in pathological measures, we performed post-hoc analyses of all PAM measures against detailed regional pathology data (totaling 95 regional measures of amyloid, tau, Lewy body, and infarct pathology in both cortical and subcortical

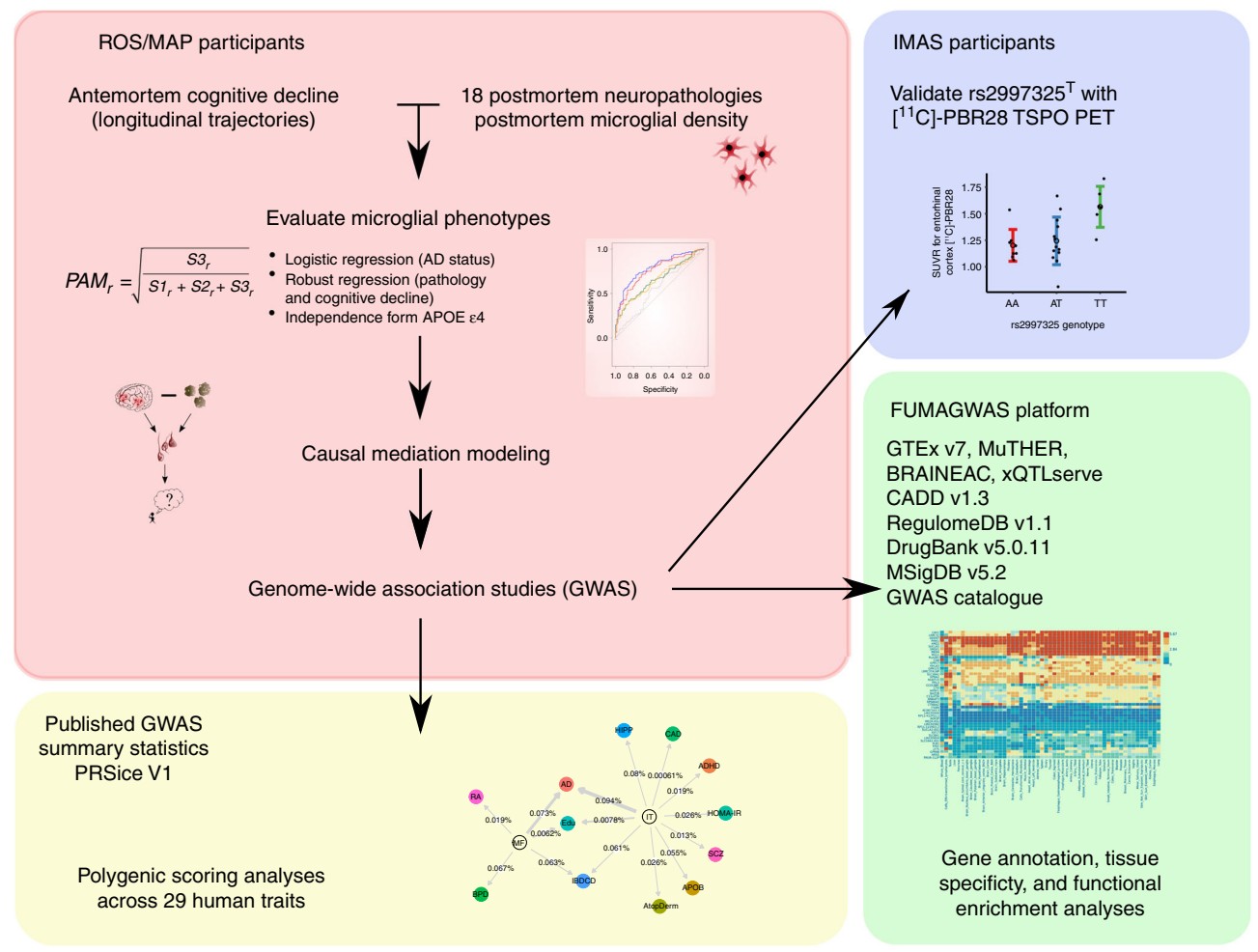

**Fig. 1** Flowchart of analyses performed in this study. ROS Religious Orders Study, MAP Memory and Aging Project, PAM proportion of activated microglia, AD Alzheimer's disease, FUMAGWAS functional mapping and annotation of genome-wide association studies, IMAS Indiana Memory and Aging Study, GTEx Genotype and Tissue Expression Study, BRAINEAC Brain eQTL Almanac, CADD combined annotation-dependent depletion, MSigDB the Molecular Signature Database. Beta amyloid figure adapted from Darvesh, Hopkins & Geula (2003) https://www.nature.com/articles/nrn1035#rightslink. Neurofibrillary tangles figure adapted from Alzheimer (1911) Ueber eigenartige Krankheitsfaelle des spaeteren Alters (10.1177/0957154×9100200506)

**Table 1 Summary statistics of ROSMAP samples included in analysis**

| Variable | MF (*n* = 225) | IT (*n* = 219) | VM (*n* = 198) | PPUT (*n* = 218) |
|---|---|---|---|---|
| Sex (F/M) | 147/78 | 141/78 | 129/69 | 140/78 |
| *APOE ε4* status (−/+) | 178/47 | 175/44 | 154/44 | 172/46 |
| PMI (mean hours, s.d.) | 8 (6.9) | 8 (6.6) | 8 (7.1) | 8 (6.9) |
| Age at study entry (mean years, s.d.) | 83 (6) | 83 (6) | 83 (6) | 83 (6) |
| Age at death (mean years, s.d.) | 89 (5.8) | 89 (5.8) | 89 (5.8) | 89 (5.8) |
| Cognitive AD diagnosis, last visit (CN/MCI/AD) | 83/64/71 | 81/61/71 | 67/58/67 | 79/61/71 |
| Postmortem AD diagnosis (AD/non-AD) | 90/135 | 86/133 | 79/119 | 86/132 |

*AD Alzheimer's disease, CN cognitively normal, F female, IT inferior temporal cortex, M male, MCI mild cognitive impairment, MF midfrontal cortex, PMI postmortem interval, PPUT posterior putamen*

regions). These analyses mirrored the brain-wide results, finding exclusively amyloid-related and tau-related associations with cortical PAM measures; there were no significant associations for either subcortical PAM measure with any regional phenotypic measure (Supplementary Figure 3).

**Mediation modeling of PAM**. We next investigated whether activated microglia might contribute to or be the result of accumulating AD pathologies. Mediation analyses found no evidence for direct or mediator effects of PAM on cognitive decline in the presence of PHFtau; rather, these analyses pointed toward indirect effects of cortical PAM on cognitive decline via PHFtau (Supplementary Data 2). Both direct and indirect effects of PAM were found for PHFtau formation, and thus the sum of evidence across our models suggests a synergistic involvement of PAM and Aβ load in affecting global cognitive decline via their effects on PHFtau (Supplementary Figure 4). Put simply, our data suggest

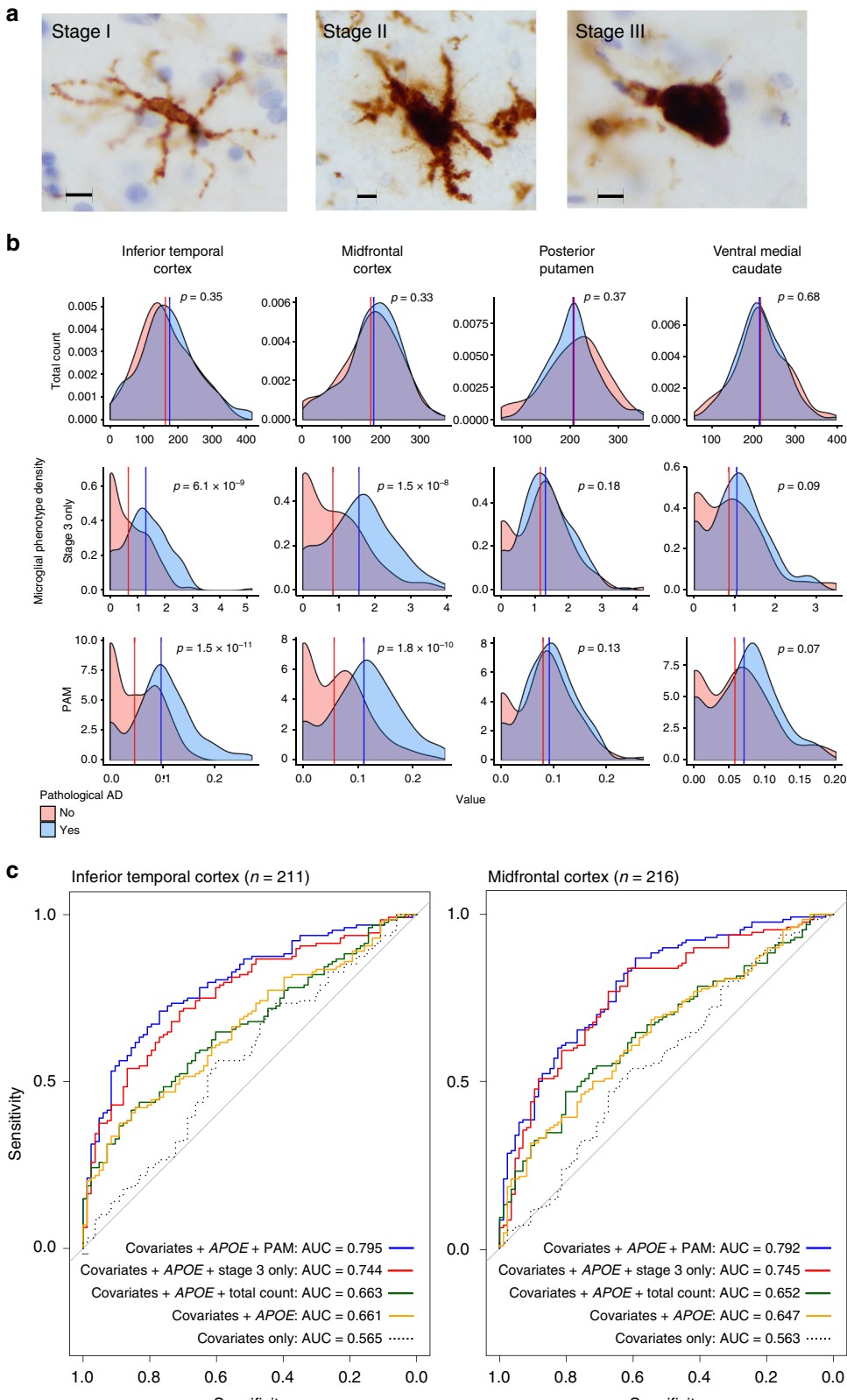

**Fig. 2** Regional distributions of microglial density phenotypes between pathological AD and non-AD. **a** Representative immunohistochemistry images of microglial staging in the MAP cohort. Scale bars are 10 μm. **b** Density plots showing the distributions of all stages of microglia (total count), active microglia (stage III only), and PAM across four brain regions ($n_{MF} = 225$, $n_{IT} = 219$, $n_{PPUT} = 198$, $n_{VM} = 218$). P-values are two-sided for Welch t-tests of means between AD and non-AD. **c** Receiver operating characteristic (ROC) curves showing model performance for cortical PAM phenotypes in logistic regression, with pathological AD diagnosis as outcome, and co-variates as specified. Area under the curve (AUC) values are for non-bootstrapped models. IT inferior temporal cortex, MF midfrontal cortex, PPUT posterior putamen, VM ventral medial caudate

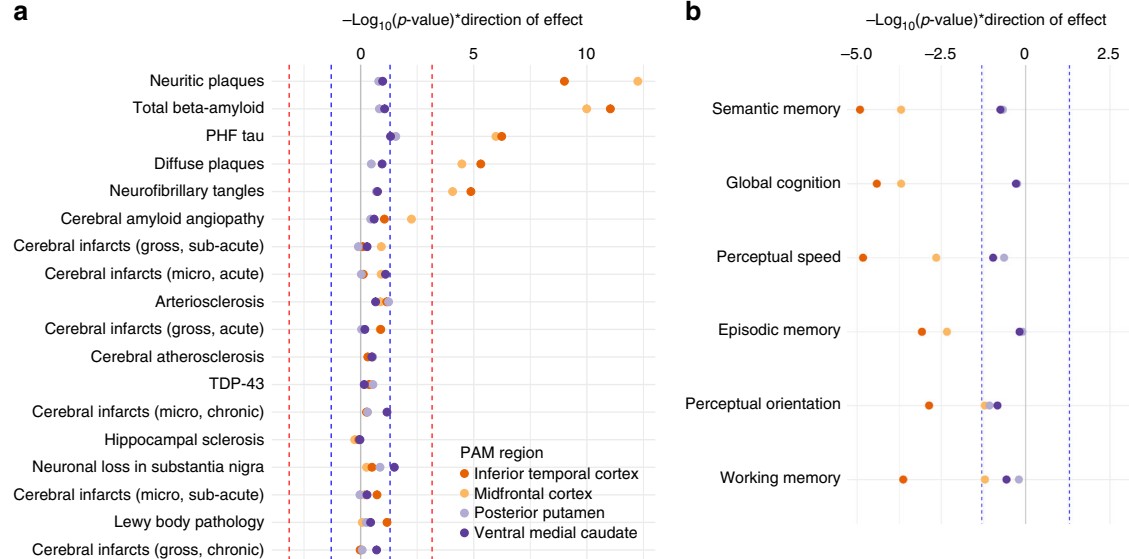

**Fig. 3** Associations of PAM phenotypes with neuropathology and cognitive decline. **a** $-\log_{10}$($p$-values), weighted by direction of effect, indicating the strength of evidence for association of each brain-wide neuropathology measure with PAM. **b** $-\log_{10}$($p$-values), weighted by direction of effect, indicating the strength of evidence for association of each measure of longitudinal cognitive decline with PAM. The red dotted lines in panel a indicate corrected statistical significance thresholds, and the blue dotted lines in both panels indicate uncorrected thresholds of $p = 0.05$. All $p$-values are two-sided and calculated from parameter estimates of iterative re-weighted least-squares regression. Model covariates included age at death, postmortem interval, *APOE* $\varepsilon4$ status, and top three EIGENSTRAT principal components ($n_{MF} = 225$, $n_{IT} = 219$, $n_{PPUT} = 198$, $n_{VM} = 218$)

the following chain of events: increased PAM → PHFtau accumulation → worsening cognitive decline. This supports and builds on cross-sectional analyses suggesting that tau correlates best with microglial activation over the course of AD[11].

**PAM and the cortical transcriptome and protein measures**. To further explore the molecular substrates of microglial activation, we accessed whole transcriptome RNA sequencing (RNAseq) and targeted proteomic data from prefrontal cortex and performed robust linear regression of each PAM measure against expression levels of 47 modules of co-expressed genes ($n = 478$ participants) and 67 proteins of interest ($n = 807$). Proteomic analyses found increases of Aβ peptide to be associated with increased MF ($p = 1.0 \times 10^{-6}$, Huber's $M = 17.2$, $n = 187$) and IT PAM ($p = 2.9 \times 10^{-5}$, Huber's $M = 15.9$, $n = 184$), providing an independent validation of our observed Aβ-PAM associations from neuropathological assessments (Supplementary Data 3). Levels of VGF were also associated with MF PAM ($p = 5.3 \times 10^{-4}$, Huber's $M = -1.8$, $n = 187$), whereby lower protein levels were observed with higher PAM (Supplementary Figure 5a). RNA module expression analyses revealed no significant PAM-module associations after correction, although the pattern of PAM-module associations differed based on region (Supplementary Figure 5b), possibly resulting from the regional specificity of the RNA sequencing data. Thus, PAM does not appear to have a strong effect on the cortical transcriptome, consistent with microglia representing just a small fraction of the total number of cells in cortical tissue.

**Genetic architecture of microglial activation**. Given the imperfect correlation between MF and IT PAM measures (Spearman $\rho = 0.66$), we performed two separate genome-wide association studies (GWAS). All significant and suggestive GWAS results are listed in Table 2 (details in Supplementary Datas 4 and 5). For IT PAM, a single locus on chromosome 1 reached genome-wide significance (rs183093970; linear regression $\beta = 0.154$, SE $= 0.024$, $p = 5.47 \times 10^{-10}$) (Fig. 4a, b). However, while

the 191 kb region encompassed by this lead SNP contains three independent signals, all three have low minor allele frequency (MAF) (0.02 > MAF > 0.015). Notably, this association was driven by only seven individuals in our sample carrying minor alleles tagging this haplotype and should therefore be considered cautiously. Beyond this genome-wide significant locus, 27 additional independent regions were associated at $p < 1 \times 10^{-5}$, and mapping based on position and combined eQTL evidence identified a total of 52 candidate genes as possible functional targets of these variants (Fig. 4c).

For MF PAM, a different locus on chromosome 1 reached genome-wide significance (top SNP: rs2997325$^T$ linear regression $\beta = 0.039$, SE $= 0.0066$, $p = 1.88 \times 10^{-8}$; Fig. 4d, e, g). Beyond this lead SNP, 11 additional regions surpassed our suggestive threshold for association, which mapped to a total of 26 genes (Fig. 4f). In contrast to rs183093970, rs2997325 is a relatively common variant (MAF = 0.37), lies 8.9 kb 3′ of a long intergenic non-coding (Linc) RNA (RP11-170N11.1, LINC01361), and influences the expression of LINC01361 in multiple tissues with combined eQTL mapping evidence of $p = 5.35 \times 10^{-12}$ [12]. However, *LINC01361* is not measured in our cortical RNAseq data. Q-value analysis revealed no significant overlap in genomic loci implicated by the two PAM GWAS, though many lead SNPs in the MF GWAS did reach nominal significance ($p < 0.05$) in the IT GWAS, and vice versa (Supplementary Data 4).

Gene enrichment analyses were performed separately on the two sets of mapped genes. For the 52 genes from the IT PAM GWAS, none of the 30 general tissue types analyzed in GTEx showed Bonferroni significant enrichment (Fig. 4j), though in more fine-grained analyses of 53 tissues, sigmoid colon was significantly enriched for differential expression of this gene set (Supplementary Figure 6b). Enrichment for functional gene categories and diseases found over 600 significant results, primarily relating to immunologic signatures (Supplementary Data 6). For the list of 26 mapped genes from the MF PAM GWAS, seven of the 30 general tissue types analyzed in GTEx showed Bonferroni significant enrichment (Fig. 4k). In contrast to

**Table 2 Independent loci identified by cortical PAM GWAS**

| PAM region | Ch | Lead SNP | SNP position | A1 | A2 | Freq (A1) | Beta (A1) | P-value | Genes mapped to locus (combined positional and/or eQTL mapping) |
|---|---|---|---|---|---|---|---|---|---|
| MF | 1 | rs2997325 | 83641424 | A | T | 0.629 | −0.0389 | 1.88E −08 | RP11-170N11.1 |
| | 1 | rs157864 | 165383761 | T | C | 0.1415 | 0.0443 | 8.60E −06 | RXRG |
| | 1 | rs651691 | 193958320 | T | C | 0.5531 | 0.0317 | 6.45E −06 | |
| | 2 | rs12623587 | 232160554 | A | C | 0.29 | −0.0321 | 6.00E −06 | C2orf72, PSMD1, HTR2B, ARMC9 |
| | 3 | rs78461316 | 104858434 | T | C | 0.0587 | 0.0749 | 2.54E −07 | |
| | 7 | rs141219652 | 70367062 | C | T | 0.0226 | 0.1083 | 8.72E −06 | |
| | 10 | rs61860520 | 134826645 | T | C | 0.0543 | −0.0719 | 4.02E −06 | TTC40, LINC01166 |
| | 11 | rs138662357 | 92058950 | C | A | 0.0555 | 0.074 | 3.49E −07 | NDUFB11P1, FAT3, PGAM1P9 |
| | 13 | rs9514523 | 106927120 | T | C | 0.1871 | −0.0386 | 9.20E −06 | |
| | 13 | rs9521336 | 110023731 | C | T | 0.2146 | 0.0378 | 1.96E −06 | MYO16-AS1, LINC00399 |
| | 14 | rs2105997 | 107209226 | T | A | 0.2276 | 0.0388 | 8.46E −06 | IGHV4-39, HOMER2P1, IGHV4-61, IGHV3-64, IGHV3-66, IGHV1-69, IGHV3-72, IGHV3-73, IGHV3-74 |
| | 15 | rs144705301 | 67855035 | C | T | 0.0263 | 0.11 | 1.74E −07 | AAGAB, RPS24P16, MAP2K5, SKOR1 |
| IT | 1 | rs56267558 | 21005316 | T | G | 0.1287 | 0.0431 | 6.40E −06 | MUL1, CDA, PINK1, PINK1-AS, DDOST, KIF17 |
| | 1 | rs113285275 | 70896319 | A | G | 0.2344 | −0.0328 | 3.10E −06 | HHLA3, CTH |
| | 1 | rs183093970 | 88454261 | G | A | 0.0147 | 0.1535 | 5.47E −10 | |
| | 1 | rs147836155 | 113501607 | T | C | 0.0129 | 0.1346 | 2.46E −06 | SLC16A1, SLC16A1-AS1 |
| | 2 | rs148259393 | 28713654 | G | C | 0.0204 | 0.1038 | 3.30E −06 | PLB1 |
| | 2 | rs141418970 | 40576095 | T | G | 0.0102 | 0.1593 | 4.82E −07 | SLC8A1 |
| | 2 | rs17018138 | 80154236 | G | C | 0.05 | 0.0588 | 7.53E −06 | CTNNA2 |
| | 2 | rs79341575 | 129133292 | C | G | 0.0551 | 0.063 | 2.75E −06 | GPR17 |
| | 2 | rs60200364 | 160708050 | A | G | 0.0623 | 0.0555 | 2.14E −06 | BAZ2B, LY75, LY75-CD302, PLA2R1, ITGB6 |
| | 2 | rs2348117 | 201598521 | T | G | 0.8893 | −0.0414 | 9.76E −06 | AOX3P, AOX2P, AC007163.3, PPIL3, RNU6-312P |
| | 3 | rs9289581 | 139405842 | T | G | 0.37 | 0.0271 | 8.32E −06 | NMNAT3 |
| | 4 | rs7656795 | 22398514 | T | C | 0.8175 | −0.037 | 1.22E −06 | GPR125 |
| | 4 | rs114105899 | 23027094 | G | A | 0.0158 | 0.1197 | 8.10E −07 | RP11-412P11.1 |
| | 4 | rs10011717 | 86136864 | A | G | 0.3774 | −0.0282 | 6.09E −06 | |
| | 4 | rs77601419 | 148249054 | T | C | 0.015 | 0.1168 | 2.22E −06 | |
| | 7 | rs77033896 | 115513816 | A | G | 0.0169 | 0.1262 | 6.84E −07 | TFEC, CAV1 |
| | 8 | rs17494322 | 20673550 | G | A | 0.075 | 0.0535 | 4.18E −06 | |
| | 11 | rs139629925 | 76144667 | G | A | 0.0121 | 0.1347 | 5.91E −06 | RP11-111M22.2, C11orf30, LRRC32 |
| | 11 | rs2084308 | 111051351 | T | A | 0.028 | 0.0967 | 4.29E −07 | |
| | 13 | rs7328235 | 41998022 | T | C | 0.9439 | −0.0607 | 6.72E −06 | MTRF1, OR7E36P |
| | 13 | rs9567982 | 48605441 | G | A | 0.1375 | 0.0394 | 3.08E −06 | LINC00444, LINC00562, SUCLA2, SUCLA2-AS1, NUDT15, MED4, MED4-AS1, POLR2KP2 |
| | 13 | rs117372720 | 61819169 | T | C | 0.0149 | 0.1092 | 8.97E −06 | |
| | 13 | rs149383020 | 112585032 | A | G | 0.0341 | 0.0787 | 5.09E −07 | |
| | 14 | rs144434563 | 91361842 | G | A | 0.0135 | 0.1265 | 6.79E −06 | RPS6KA5 |
| | 14 | rs137899216 | 91830706 | T | C | 0.0277 | 0.0911 | 3.06E −06 | GPR68, CCDC88C, SMEK1 |
| | 17 | rs112645358 | 66248531 | T | C | 0.0384 | 0.069 | 9.94E −06 | KPNA2, LRRC37A16P, AMZ2, ARSG |
| | 18 | rs148222222 | 65675614 | T | C | 0.0169 | 0.1133 | 1.47E −06 | RP11-638L3.1 |
| | 20 | rs71336998 | 5467150 | G | A | 0.018 | 0.1266 | 1.17E −07 | LINC00654 |

*Ch* chromosome, *A1* allele 1 (effect allele), *A2* allele 2, *eQTL* expression quantitative trait loci, *Freq* allele frequency, *IT* inferior temporal cortex, *MF* midfrontal cortex

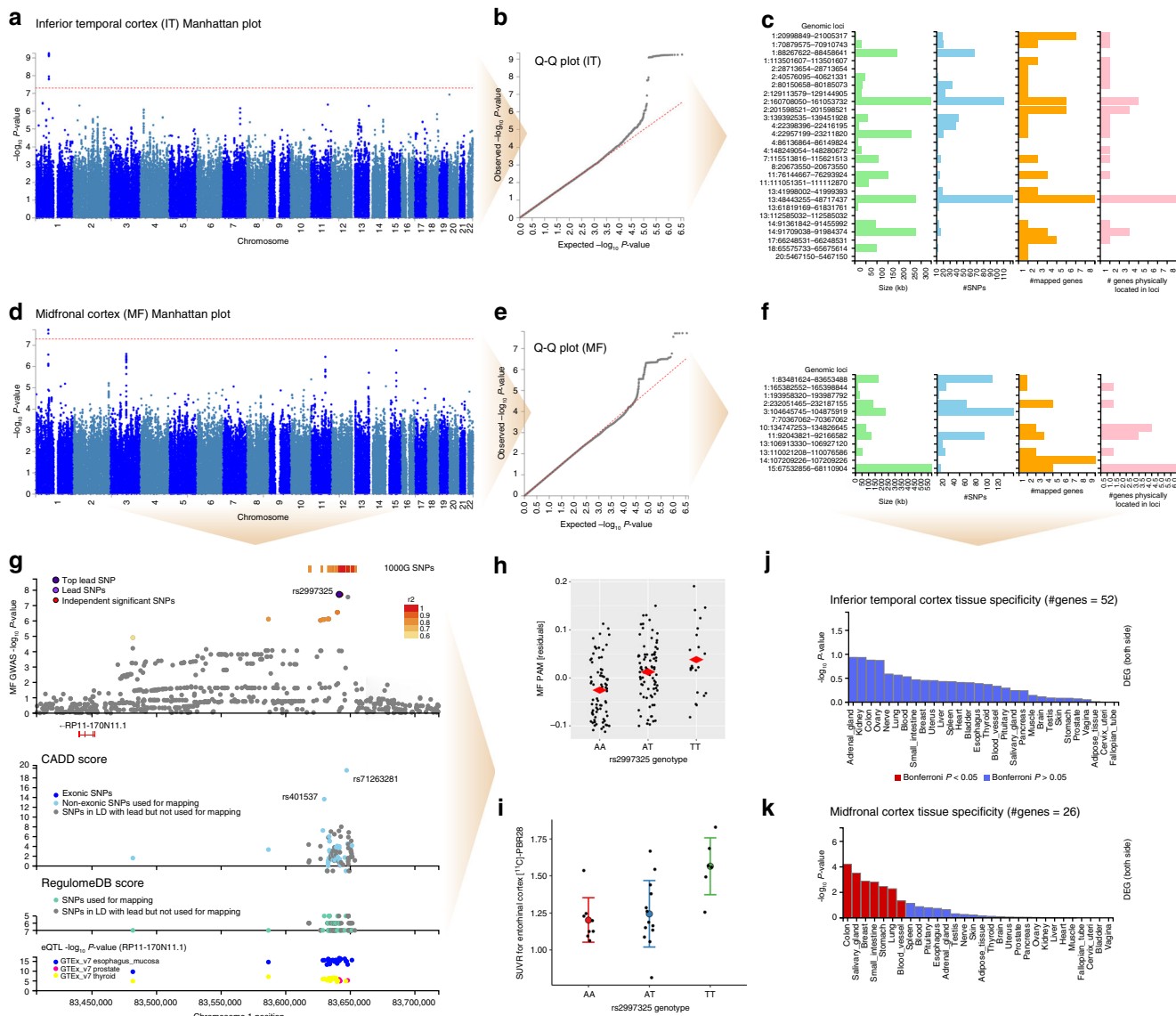

**Fig. 4** Genome-wide association studies (GWAS) of PAM with TSPO PET imaging follow-up. **a** Manhattan plot, (**b**) Q–Q plot, and (**c**) locus summary chart for inferior temporal cortex (IT) PAM, and corresponding, (**d**) Manhattan plot, (**e**) Q–Q plot, and (**f**) locus summary chart for midfrontal cortex (MF) PAM GWAS. Both analyses co-varied for genotype platform, age at death, sex, postmortem interval, *APOE* ε4 status, and top three EIGENSTRAT principal components. **g** Regional association plot highlighting the MF PAM genome-wide significant locus surrounding rs2997325 ($p = 1.88 \times 10^{-8}$, $n = 225$). The color of each dot represents degree of linkage disequilibrium at that SNP based on 1000 Genomes Phase 3 reference data. The combined annotation-dependent depletion (CADD) score is plotted below the regional association plot on a scale from 0 to 20, where 20 indicates a variant with highest predicted deleteriousness. The RegulomeDB score is plotted below the CADD score, and summarizes evidence for effects on regulatory elements of each plotted SNP (5 = transcription factor-binding site or DNAase peak, 6 = other motif altered, 7 = no evidence). Below the RegulomeDB plot is an eQTL plot showing $-\log_{10}(p\text{-values})$ for association of each plotted SNP with the expression of the mapped gene RP11-170N11.1 (*LINC01361*). **h** Strip chart showing the significant relationship between MF PAM (on *y*-axis as GWAS covariate-only model residuals) and rs2997325 genotype. **i** Whisker plot showing the means and standard deviations of [$^{11}$C]-PBR28 standard uptake value ratio (SUVR) for the left entorhinal cortex in the PET imaging sample from the Indiana Memory and Aging Study ($p = 0.02$, $r^2 = 17.1$, $n = 27$), stratified by rs2997325 genotype. Model co-varied for *TSPO* rs6971 genotype, *APOE* ε4 status, age at study entry, and sex. Tissue enrichment analyses for (**j**) IT and (**k**) MF PAM gene sets in 30 general tissue types from GTEx v7 show Bonferroni significant enrichment (two-sided) of only the MF gene set with colon, salivary gland, breast, small intestine, stomach, lung, and blood vessel tissues. Heat maps showing enrichment for all 53 tissue types in GTEx v7, including uni-directional analyses for up-regulation and down-regulation specifically can be found in Supplementary Figure 5

functional enrichment analyses for the 52 IT PAM GWAS genes, a total of only six unique gene sets showed significant enrichment for MF PAM genes (Supplementary Data 6). Importantly, among the 78 genes mapped between both GWAS, eight encoded proteins targeted by known drugs (Supplementary Data 8). Of these, some, such as Acitretin and Eletripan, are relevant to

immunological and neurological illness; they are used in the treatment of psoriasis and migraine, respectively.

Given the absence of available replication datasets with postmortem microglial staging and genome-wide genotype data, we pursued confirmatory analyses of our GWAS results using another measure of microglial activation: in vivo translocator

protein (TSPO) positron emission tomography (PET) imaging using the [11C]-PBR28 radioligand. Due to the low MAF of rs183093970, we could only test the genome-wide significant variant from our MF PAM GWAS (rs2997325). In this slightly younger sample ($\mu_{age} = 71.2$ years), we found that rs2997325[T] was significantly associated with an increase in [11C]-PBR28 binding in the left entorhinal cortex in vivo ($p = 0.02$, $r^2 = 17.1$; Fig. 4i), consistent with our finding that the same allele increased MF PAM (Fig. 4h).

**Role of activated microglia across human traits.** Having generated genome-wide profiles of genetic risk for both cortical PAM measures, we used a high-resolution polygenic scoring-based method to test for overlap in the genomic underpinnings of microglial activation and AD, which would suggest a causal link between microglial activation and AD susceptibility. Secondarily, we tested 28 other brain and immune-related traits with publicly available GWAS data to more broadly assess the role of microglia in susceptibility to human disease. For MF PAM (Fig. 5a, b), AD showed the strongest polygenic association ($p = 1.8 \times 10^{-10}$, $r^2 = 7.3 \times 10^{-4}$), and five other traits had optimal evidence for co-genetic regulation at a corrected threshold, with educational attainment showing the second strongest effect ($p = 1.1 \times 10^{-5}$, $r^2 = 6.2 \times 10^{-5}$). For IT PAM, AD susceptibility ($p = 4.9 \times 10^{-13}$, $r^2 = 9.4 \times 10^{-4}$) was also the most strongly associated, and 12 other traits met our significance threshold with educational attainment ($p = 8.2 \times 10^{-7}$, $r^2 = 7.8 \times 10^{-5}$) also demonstrating the second strongest effect. These analyses provide evidence that the genetic predisposition to having activated microglia also contributes to making an individual more likely to develop AD. This result therefore goes beyond simply finding an enrichment of AD genes that happen to be expressed in microglia: they provide evidence that genomic propensity for the active microglia state is causally related to AD risk.

The association with educational attainment is intriguing and consistent with our understanding that microglia play an important role in sculpting the developing brain by pruning synapses[13]. Likewise, the association of IT PAM with schizophrenia extends the narrative of the involvement of microglia in this neuropsychiatric disease[14]. To be thorough, we repeated these analyses in the reverse direction, asking whether genetic susceptibility for each of these 29 traits influences microglial activation. In these analyses, we found no association for AD, suggesting that the causal chain of events most likely flows from genetic risk → microglial activation → AD. We did find that primary sclerosing cholangitis (PSC) risk influenced both PAM measures while a few other traits demonstrated single associations (Fig. 5c, d). The PSC and inflammatory bowel disease associations may represent shared architecture relating to activated myeloid cells, since microglia and peripheral macrophages (which are implicated in these two inflammatory diseases) share many molecular functions.

## Discussion

Microglial activation is a well-known phenomenon that has been implicated in a myriad of pathological processes. However, a majority of studies have been conducted either in small numbers of human subjects with limited clinicopathologic data or in murine model systems whose relevance to human disease is unclear given (1) that they do not recapitulate the events and conditions observed in the aging human brain and (2) the emerging understanding of significant differences between aging human and murine microglia[15,16]. Our study provides several important insights. First, microglial activation is not a general feature of the AD brain: we found it to be elevated in cortical but

not in subcortical structures. Further work now needs to be conducted to sample a wider array of regions in individuals with a range of pathologic burden and those without pathology so that we can better understand the extent to which these regional differences may be related to the nature of the brain region itself or the extent to which that region is affected by amyloid or tau pathology in a given individual. Currently, the data are too sparse to be able to interpret our results beyond simply noting that PAM in subcortical regions do not appear to relate to the burden of AD pathology. Second, other common neuropathologies of older age do not appear to be significantly associated with activated microglia in the regions tested and therefore do not confound observed associations with Aβ and tau pathologies. Third, our modeling suggests that microglial activation leads to cognitive decline indirectly via the accumulation of PHFtau. This histology-based result is consistent with transcriptome-based findings we have recently reported[17].

We also describe the previously unmapped genetic landscape of microglial activation measured in postmortem human brain; we found no effect of the *APOE* locus despite reports that APOE may be a ligand for important microglial receptors such as TREM2[18,19]. However, a discovery GWAS returned two significant results: the uncommon rs183093970 variant, whose role remains to be validated, and rs2997325, whose role we confirmed by assessing the related phenotype of in vivo microglial PET imaging. The effect of the rs2997325[T] allele in increasing microglial activation and [11C]-PBR28 binding should be validated more extensively, but, given its high frequency and strong effect, it may be a clinically relevant biomarker requiring attention in the many studies evaluating the utility of other microglia-targeted ligands in a clinical setting to diagnose or stage neurological diseases. While little is known about the biology of *LINC01362* that appears to be influenced by rs2997325, several of the suggestive loci implicated by our PAM GWAS were found within or near genes of functional significance. For example, the Lymphocyte antigen 75 gene (*LY75*), codes for CD205, a dendritic cell surface receptor that interacts with MHC class I molecules[20] and plays an important role in T cell function[21]. Further investigation of cortical PAM genetic architecture is warranted to extend our initial set of observations; replication of associations with both histologic measures of microglial activation and TSPO imaging in populations at risk of AD are needed to assess the utility of these measures as biomarkers that may inform the diagnosis and/or management of aging-related cognitive decline. In addition, in vitro knock-down and DNA editing experiments, combined with amyloid and tau-sensitive assays, will be needed to explore the role of these variants in biological events related to AD.

There are important limitations to consider in our study. First, while we were able to test for some regional-specific effects of PAM on pathology, we did not always have data on pathology in the exact tissue samples in which microglia were counted. Therefore, tightly coupled pathology–microglial associations, such as those known to exist with acute infarction[22], may have been missed. Whether or not microglial activation is a region-specific phenomenon in aging is an unresolved question; both global and focal distributions have been reported in the aging brain, albeit using different measurements[23,24]. Reassuringly, a recent postmortem investigation of microglial activation measured morphologically in the brains of 11 late onset AD subjects vs. 12 age-matched controls also found increases in microglial activation in cortical but not sub-cortical regions[4]. Second, we have a limited sample size for genome-wide analyses, although it was sufficient to discover the rs2997325 variant which has a strong effect on both microglial activation and in vivo microglial imaging. Third, our moderate sample size also means that we

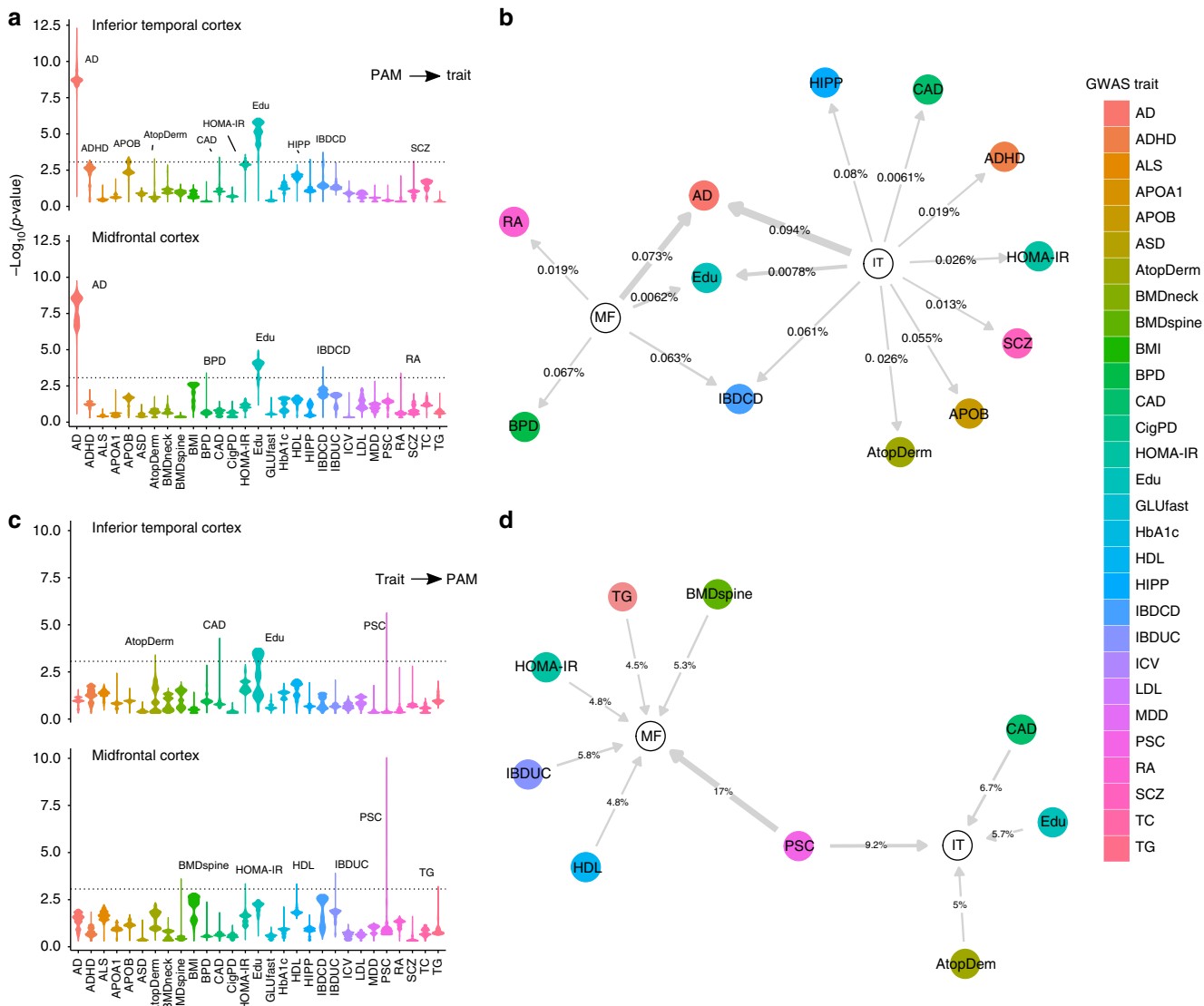

**Fig. 5** Results of polygenic scoring analysis of PAM GWAS and 29 published traits. **a** Stratified violin plots for PAM → published trait polygenic analyses, showing $-\log_{10}$(p-values) for genetic score (GRS) associations of inferior temporal cortex (IT) PAM and midfrontal cortex (MF) PAM across all p-value thresholds ranging from 0 to 0.5, with $5.0 \times 10^{-5}$ regular increments (10,000 total scores). Each PAM genetic score was mapped onto the summary statistics of each published GWAS trait and tested for significance. The dotted line represents a corrected statistical significance threshold of $p = 8.6 \times 10^{-4}$, corrected for 29 GWAS traits and two PAM measures. The width of each violin represents the density of PAM polygenic scores associated with each trait at a given significance. For example, a bottom-skewed violin (e.g. educational attainment (Edu)) indicates that a majority of scores across the tested set of thresholds tended to achieve greater significance, whereas a top-skewed violin (e.g. bipolar disorder (BPD)) indicates that a majority of tested scores tended toward lower significance. Peaks achieving at least one score above corrected statistical significance are labeled for their respective GWAS trait. **b** Network plot illustrating significant results for PAM → GWAS trait analyses, where an arrow indicates a directional effect of a peak PAM GRS on a GWAS trait at corrected significance (edges are labeled with % variance explained in GWAS trait by optimal PAM GRS, and edge thickness is proportional to the $-\log$(p-value) of the association). **c** Stratified violin plots for the GWAS trait → PAM analyses, such that GRS were first calculated across thresholds for published traits and then tested against the PAM summary statistics. **d** Network plot illustrating significant results of the GWAS trait → PAM analyses. GRS genetic risk score, AD Alzheimer's disease, Edu educational attainment, IBDCD irritable bowel disease, Crohn's disease; IBDUC irritable bowel disease, ulcerative colitis, CAD coronary artery disease, HIPP hippocampal volume (from MRI), RA rheumatoid arthritis, BPD bipolar disorder, MDD major depressive disorder, HOMA homeostasis model assessment (from fasting insulin and glucose); AtopDerm atopic dermatitis, APOB circulating apolipoprotein B, HDL high-density lipoprotein cholesterol, BMDneck bone marrow density of the neck, SCZ schizophrenia, ADHD attention deficit/hyperactivity disorder. The full list of abbreviations found in the legend and descriptions of each published GWAS, including sample sizes, are listed in Supplementary Data 7

cannot exclude the possibility that activated microglia may have weak, undetected effects on non-AD pathologies. Finally, we acknowledge that our chosen methodologies for quantifying microglia have intrinsic biases and limitations. For example, the manual identification of stained microglia carries a degree of subjectivity and, while [$^{11}$C]-PBR28 is a well-validated biomarker

of microglial activation in humans, experimental confounds and the potential for non-specific binding are important to consider in any molecular imaging experiment. Further, we recognize that infiltrating macrophages present in brain tissue samples may represent a source of error in our counts since they are largely indistinguishable from microglia once they are activated.

Reassuringly, recent single-cell analyses performed by our group have shown that these cells represent a negligible minority (~1%) of myeloid lineage cells present in brain tissue from similar subjects[25]. These same single-cell sequencing experiments will be important for identifying molecular subpopulations of microglia that may correspond to those activation states captured by the PAM phenotype.

Despite these limitations, our polygenic analyses revealed significant links that clarify and expand the roles of activated microglia in human diseases. This is clearest for AD where we refine the narrative of the involvement of myeloid cells in AD susceptibility by showing that the proportion of activated microglia has a causal role in AD. A similar result for cognitive attainment is intriguing as it may be capturing microglia's role in synaptic pruning during development and memory consolidation; however, this is confounded by the fact that this enrichment may also be capturing the role of microglia in age-related cognitive decline that we describe in this report. The enrichment for Crohn's disease is unlikely to represent a causal role of microglia in this disease; rather, this result captures the likely overlap of genetic architecture between activated microglia and other activated myeloid cells in the periphery that are known to be involved in Crohn's disease.

The relevance of recent reports of pathologically important subtypes of microglia in murine model systems, such as disease-associated microglia (DAM)[1] and dark microglia[26], remains to be demonstrated in the aging human brain since mouse and human microglia diverge significantly in functional molecular signatures with age[27]. By studying activated microglia in the target organ in humans, we have found several key insights, including (1) the observation that they may accelerate cognitive decline through an effect on PHFtau accumulation, which enables us to design mechanistic studies, (2) the discovery and validation of a chromosome 1 locus (among several other suggestive plausible loci) that provides a biological foundation for dissecting the mechanisms of microglial activation, and (3) the effect of rs2997325 on [11C]-PBR28 binding which may need to be accounted for in human imaging studies of in vivo microglial activation. Thus, microglial activation is a central component of AD susceptibility, and we have begun to elaborate its place in the causal chain of events leading to increased accumulation of tau pathology and subsequent cognitive decline, as well as regulatory mechanisms that influence this activation.

## Methods

**Study subjects**. All antemortem cognitive and postmortem data analyzed in this study were gathered as part of the Religious Orders Study and Memory and Aging Project (ROS/MAP)[28–30], two longitudinal cohort studies of the elderly, one from across the United States and the other from the greater Chicago area. All subjects were recruited free of dementia (mean age at entry = 78 ± 8.7 (SD) years), agreed to annual clinical and neurocognitive evaluation, and signed an Anatomical Gift Act allowing for brain autopsy at time of death. In vivo [11C]-PBR28 PET imaging acquisitions were collected on data from the Indiana Memory and Aging Study (IMAS), an ongoing neuroimaging and biomarker study based at the Indiana University School of Medicine including elderly subjects at multiple levels of cognitive impairment[31]. Written informed consent was obtained from all ROS/MAP and IMAS participants. Study protocols were approved by each site's Institutional Review Board. Full methods can be found in Supplementary Methods.

**Genomics**. Genotype array data for 2067 ROS/MAP subjects were imputed using the Michigan Imputation Server (Haplotype Reference Consortium reference v1.1). For IMAS, data were imputed using IMPUTE v2.2 (1000 Genomes phase 1 reference). *APOE* (rs429358 and rs7412) genotyping was carried out separately using standard protocols.

**Tanscriptomics**. RNA sequencing of postmortem dorsolateral prefrontal cortical tissue from 538 ROS/MAP subjects were available for analysis at the time of study. Following sequencing and standard quality control[32], The Speakeasy algorithm[33] was used to cluster expressed genes into functionally cohesive gene modules, which have been extensively validated for robustness and pathophysiological relevance[16,34]. Average values of gene expression for each of 47 modules were used as quantitative outcomes.

**Proteomics**. Selected reaction monitoring-based (SRM) quantitative proteomics was used to measure the abundance of 67 proteins (Supplementary Data 3) in frozen dorsolateral prefrontal cortical tissue from 807 ROS/MAP participants according to a standard protocol[35,36].

**Postmortem neuropathology**. A total of 18 disease-related and age-related neuropathologies were measured brain-wide ($n = 985$). Pathologies included multiple validated measures of Aβ peptides, neuritic and diffuse plaques, hyperphosphorylated tau protein, neurofibrillary tangles, micro and macro cerebral infarcts, cerebral atherosclerosis, TDP43 proteinopathy, and hippocampal sclerosis, among others (see Supplementary Methods)[37]. A subset of up to 225 brain samples were also evaluated for the presence of microglia at three stages of activation in four regions (midfrontal (MF) cortex, inferior temporal (IT) cortex, ventral medial caudate (VM), and posterior putamen (PPUT)), based on morphology: stage I (thin ramified processes), stage II (plump cytoplasm and thicker processes), and stage III (appearance of macrophages).

**Antemortem cognitive decline**. A total of 1932 subjects with genomic data also had longitudinal cognitive performance data available at the time of study (the full list of cognitive tests can be found in Supplementary Data 10).

**In vivo TSPO PET imaging**. Scans of in vivo [11C]-PBR28 binding were assessed in 27 subjects ($n_{CN} = 13$, $n_{MCI} = 7$, $n_{AD} = 7$) as part of the IMAS study[31].

**Statistical analysis**. Regression analyses were performed in R (v3.3.3)[38]. PAM was calculated by the following formula:

$$\mathrm{PAM}_r = \sqrt{\frac{S3_r}{S1_r + S2_r + S3_r}} \qquad (1)$$

where $r$ represents each of four regions and S1, S2, and S3 represent microglial densities measured in region $r$ at stage I, II, and III, respectively. To address potential concerns related to PAM distributions and model fitting, we performed extensive model validation and sensitivity analyses (Supplementary Methods; Supplementary Figure 2; Supplementary Data 1). Differences in model performance are reported as change in C-index, expressed as a percent, which corresponds to the area under the receiver operating characteristics curve (ΔAUC). Iterative re-weighted least-squares robust regression was used and model validation was performed using the .632+ bootstrap method[39]. Models were corrected using the Bonferroni procedure. Causal mediation modeling for identifying direct and indirect effects of PAM on cognitive decline was performed using the 'mediation' R package.

GWAS were performed in PLINK[40] (v1.90) using imputed genotype dosages and additive linear models, co-varying for age at death, postmortem interval (PMI), sex, genotype batch, and the first three EIGENSTRAT[41] principal components. Significance thresholds of $p < 2.5 \times 10^{-8}$ and $p < 1.0 \times 10^{-5}$ (two-sided) were deemed genome-wide significant and suggestive, respectfully. Post-processing of GWAS results was conducted using the full complement of state-of-the-art tools available through the recently released Functional Mapping and Annotation of Genome-Wide Association Studies platform (FUMAGWAS; http://fuma.ctglab.nl/)[42]. Analyses included positional and eQTL-based gene mapping, assessment of Combined Annotation Dependent Depletion (CADD v1.3)[43] and RegulomeDB scores[44], tissue expression specificity using GTEx v7, comprehensive gene set enrichment analyses, and mapping of gene targets to the DrugBank database[45].

Statistical Parametric Mapping version 8 (SPM8) was used for imaging analysis. Freesurfer (v5.1) was used to define subject-specific regions of interest and average standardized uptake value ratios (SUVRs) were extracted for three bilateral ROIs. As six ROI values were strongly correlated and the sample size is moderate for genetic analysis of this type ($n = 27$), we carried out a multivariate analysis by performing genetic association analysis using six phenotypes simultaneously to minimize the number of test performed and increase the statistical power. We used the software for correlated phenotype analysis (SCOPA) program[46] to perform a single omnibus genetic test for all correlated phenotypes, modeling rs2997325 genotype additively.

To assess causal relationships between the genetic determinants of cortical PAM (based on our PAM GWAS results) and 29 brain and immune-related traits (and vice versa), we used a genetic risk score-based method[47] as implemented in the PRSice[48] program (v1.25). The full list of traits and published GWAS references are listed in Supplementary Data 7. Bonferroni correction was applied for 29 traits in both directions, resulting in a significance threshold of $p < 8.6 \times 10^{-4}$.

**Reporting summary**. Further information on experimental design is available in the Nature Research Reporting Summary linked to this article.

## Data availability

Access to ROS/MAP data used in the preparation of this manuscript can be applied for at the Rush Alzheimer's Disease Center Resource Sharing Hub and is stored on Synapse Accelerating Medicines Partnership—Alzheimer's Disease (AMP-AD) Knowledge Portal (https://www.synapse.org/#!Synapse:syn3219045) [10.7303/syn3219045]. A reporting summary for this Article is available as a Supplementary Information File.

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

## Acknowledgements

We would like to thank all of the study participants and acknowledge the essential contributions of Chaya Gopin and Kimberly Cameron to the recruitment and clinical assessments of those participants. We are indebted to the participants in the Religious Orders Study and the Rush Memory and Aging Project. We thank the staff of the Rush Alzheimer's Disease Center. Work was supported by NIH grants P30AG10161, R01AG15819, R01AG17917, R01AG30146, R01NS084965, R01AG048015, U01AG046152, R01LM012535, R03AG054936, the Illinois Department of Public Health, the Translational Genomics Research Institute, and a Postdoctoral Fellowship from the Canadian Institutes of Health Research (CIHR).

## Author contributions

D.F. was responsible for study design, data management and pre-processing, ROS/MAP statistical analyses, and writing of the manuscript. T.R. and E.M.B. contributed to the study design, ROS/MAP statistical analyses, and editing of the final manuscript. S.L.R. and K.N. were responsible for processing IMAS PET imaging data and performing all imaging analyses. V.P. was responsible for the SRM proteomic methodology and analysis, and assisted in editing of the final manuscript. J.A.S. was responsible for overseeing neuropathological data acquisition, ensuring quality control of the data, and editing the final manuscript. A.S. was responsible for IMAS study design and acquisition of IMAS data, and manuscript editing. D.A.B. oversees the ROS/MAP studies, contributed to study design, and assisted in manuscript editing. P.L.D.J. contributed to the study design, evaluation of results, and writing of the manuscript. All authors read and approved the final manuscript.

## Additional information

**Competing interests:** The authors declare no competing interests.

