## [Peer Review File · Nature Communications]

Reviewers' comments:

Reviewer #1 (Remarks to the Author):

Felski et al. show that the proportion of activated microglia (PAM), quantified by morphological analysis in patient brains of the ROS/MAP study, is more discriminative of Alzheimer's disease (AD) status than other tested parameters. PAM improves logistic regression model performance compared to APOE ϵ 4 and is associated with histological hallmarks of AD e.g. total A β load and others. Moreover, the authors show that an increase in PAM parallels an increase in AD related pathological features as well as cognitive decline and is not associated with other age related pathologies. The authors claim their work confirms the higher importance of microglia activation over microglia proliferation in AD pathogenesis and a synergistic role of PAM and A β load via effects of PHF Tau to promote AD pathology. In follow-up high throughput analyses the authors show increased PAM to be associated with increased A β peptide, CIQA, DOSK2 and decreased VGF via proteomics, yet found no significant associations between PAM and transcriptome changes measured via RNA Sequencing. Using GWAS, the authors claim association of IT PAM with rs183093970 and show association of MF PAM with rs2997325T and claim it could be a new biomarker. LINC01361 was not affected by rs2997325T. The authors could show an association between PAM-linked genes (based on GWAS) and immunological signatures as well as known drug targets, yet could not confirm association between rs2997325T and microglia activation using TSPO PET. Finally, the authors claim that a genetic predisposition to activated microglia is a cause of AD pathogenesis and is reflected by their polygenic scoring-based methods.

The study is of importance to the microglia research community. It represents one of the largest studies analyzing microglia activation state in humans in the context of Alzheimer's disease and ageing so far. The work will influence thinking in the field, as it seems to confirm, yet with only one model, that both microglia activation and Amyloid β accumulation contribute to Tau pathology and eventually to cognitive decline.

Major critiques

- unclear description how the microglia were visualized and how many samples were analyzed
- unclear reasons of choice of the particular way to detect microglia activation in brain samples
- no histological evidence that activated microglia and A β synergistically lead to cognitive decline via PHF Tau
- no mentioning of sample number for all high throughput analyses
- masking of microglia-specific transcriptome differences associated to PAM by application of bulk RNA Sequencing
- differences found for VGF, CIQA and DOCK2 were not followed up
- no differences in LINC01361 expression in the presence of rs2997325T
- no confirmation of validity of rs183093970 and rs2997325T by TSPO PET
- study seems overloaded by the high throughput data mentioned in a descriptive fashion
- the last part of the results is too descriptive
- putative bias by lack of information about pathological data of tested patients should be mentioned in the results part and ruled out by respectively reanalyzing the data

Minor critiques

- the number of samples analyzed are not mentioned in the figure legends
- figure 4 g and i are not explained
- the results part is hard to read as it is too condensed in particular the first part of the results
- discussion about putative bias through the applied quantification method for microglia activation and previous studies using this method including TSPO-PET is missing

- outlook on how rs183093970 and rs2997325T could be further confirmed as new biomarkers is missing

In conclusion, the authors show the association between the PAM score and pathological features of AD as well as cognitive decline. The statistical approaches and models used seem valid and appropriate. Covariates such as age at death, postmortem interval, APOEε4 status, and principal components are included in the described models.

Yet, it is necessary to rule out a putative bias caused by lack of information about patient pathological data that the authors mention in the discussion. The authors need to confirm their results by applying a further method to visualize microglia activation in order to rule out the respective bias of their method. In addition, the results of the proteomics, RNA- Sequencing and GWAS analyses should be followed up and confirmed further. The last part of the results can be left out to simplify the manuscript.

The manuscript should be considered for publication after including the following experiments to support the claims of the authors:

- reanalysis of the PAM associations excluding patient samples with incomplete information on pathological data
- reanalysis of PAM scores using a further method to analyze microglia activation e.g. Cd68, TMEM119 IHC
- quantification of protein levels of VGF, CIQA and DOCK2 in patient samples to confirm association with PAM
- if possible TSPO PET with more patients to confirm validity of rs2997325T and rs183093970 as biomarkers
- in situ hybridization-based quantification of LINC01361 in patient brain samples

Reviewer #2 (Remarks to the Author):

This is an interesting paper that uses several analyses to investigate PAM (proportion of activated microglia) to neuropathology, genetics, and AD status.

Main comments:

1. I found the methods a little complicated and think that a flow diagram would be really helpful.
2. Why was a GWAS threshold of 2.5×10^{-8} used rather than a more liberal (but more widely applied) 5×10^{-8} threshold?
3. Line 178: I don't think it's meaningful to describe an e.g., 18% improvement in model performance when considering AUCs. A better metric would be sensitivity, specificity, PPV and NPV.
4. Line 182: for the comparison with APOE, what was the point estimate for ε4ε4 carriers? I appreciate that there won't be too many in the study but it would be interesting to report this. Also, was ε4 status associated with pathology?
5. Line 198: What was the statistical model used for the cognitive decline analysis? On line 201 I find it strange that there is 'no sig effects of subcortical PAM on cognitive decline' when PAM measurement was taken after the cognitive data.

6. Paragraph from line 202: I think that the conclusions for this section are too strong. Given that PAM and pathology were measured at death, I don't think the authors can justify a PAM -> pathology -> cog decline pathway. Given the nature of the data collection, I don't think directionality can be assessed here.

7. Line 222: I don't find the nominal (non-significant) reporting to be helpful unless there is external replication.

8. Line 238: I am not convinced by the rare variant finding. A MAF <1.5% in such a small cohort does not seem appropriate as outliers will drive associations at this threshold. I think that a more strict MAF and MAC should be applied here.

9. Line 250: What was the genetic correlation between the two PAM GWAS?

10. Line 264: are any of these drugs relevant?

11. Line 272: how many statistical tests were performed here?

12. Paragraph from Line 275: the predicted r^2 values here are extremely small <0.1%. What sort of effect sizes are observed if you simulate under the null?

13. From line 292: as with previous comments, I wonder how the timing of the PAM measure (at post-mortem) will impact interpretation in causal modelling. I think this is complicated and needs a caveat.

14. The ROS data are from "across the United States" but site information does not seem to have been included in the modelling. Is the random intercept term for site non-zero in a model with PAM as the outcome?

Minor comments:

Line 54: typo "a molecularly defined subtypes"

Line 66: I am not a fan of using the word 'unique'. I don't think it adds anything here.

Line 82: of "the" elderly?

Line 90: typo "Tanscriptomics"

Line 106: why were these regions selected?

Line 110: it would be good to know the list of 17 cognitive tests that were used. This info is not present in the supplement.

Line 125: typo co-varying

Line 126: PMI - please spell out

Line 129: please list the suite of analyses that were performed using FUMA. 'The full complement of state-of-the-art tools' is not clear.

Line 154: typo (S)pearman

Line 158: please state the n's (or n range) for the much smaller studies

Line 160: Why were Welch t-tests used? The data don't look terribly Gaussian. I think Mann-Whitney tests might be more appropriate although this shouldn't change the results.

Line 193: PHF - please spell out

Line 193: There are 18 pathologies in Figure 2a but 14 reported on line 191. They're explained more fully in the supplementary methods but having a sentence about what the measured neuropathologies were somewhere in the main text might be helpful

Line 194: 'increase' suggests that changes in PAM have been observed. Maybe replace with 'higher'

Line 247: 8.9 kb 3' - is there something missing here?

Line 294: Please add a ref to Figure 5 as this is the first mention of SCZ

Line 313: It might be interesting to expand this a bit and relate it to the differences in cortical/subcortical pathology

Reviewer #3 (Remarks to the Author):

Neuropathological correlates and genetic architecture of microglial activation in elderly human brain

This paper has examined the microglial morphology in the elderly and compared the proportion of activated microglia found in the brain with markers of Alzheimer's-related dementia (i.e. amyloid- β , tau and rate of cognitive decline). The authors also performed genome-wide analysis to identify a common variant (rs2997325) that impacts microglial activation. The authors of this study have primarily used mathematical modeling to investigate and interpret their findings. This is an interesting paper, but would benefit from additional figures and descriptions.

1) The microglia are divided into three activation stages; stage I (thin ramified processes), stage II (plump cytoplasm and thicker processes), and stage III (appearance of macrophages). This reviewer finds it very surprising that we never see an image from the histological analysis to demonstrate these three subgroups. There is no information on the staining protocol or the stain(s) used on the slices, the analysis protocol is completely missing (how many sections were analysed per individual or how many cells), the factors used to define the processes as thick or thin is absent (was this a program? By eye? Based on size i.e. nm etc?). We see no quantification of the histology results. This is all required for the paper as these findings are the basis on which the entire mathematical modeling is founded.

2) The third stage of microglia activation is also troubling to this reviewer, i.e. the "appearance of macrophages". As we have no idea what stain or antibodies are used to define macrophages in the brain, one can only imagine how this cell population was distinguished from the resident microglial cells. It is very difficult to distinguish microglia from macrophages, as they are both cells of the myeloid lineage they share many similar receptors and markers. How were the macrophages removed from the analysis to only focus on microglia? This is a critical point that needs to be addressed with suitable example pictures for the paper.

3) There are a number of grammatical errors throughout the manuscript, and abbreviations are sometimes used without introducing them first. The sentence running from page 13 to 14 is incomplete.

4) Page 9 sentence 171, 172 states microglia activation rather than proliferation mediates neurodegeneration but the paper referenced is from 2002 and since then there have been other reports demonstrating the microglia proliferation does have a role in neurodegenerative disease development, e.g. Gomez-Nicola et al 2013 "Regulation of Microglial Proliferation during Chronic Neurodegeneration". This should be updated in the text.

5) It is not clear what cognitive tests were performed on participants to determine cognitive decline, and AD diagnosis. A table should be included.

6) It would be beneficial to see the quantification of tau and amyloid on the different disease groups, before being used in the regression studies. This could also be included as a table.

7) The authors use "educational attainment" as an effect in their modeling, yet they do not provide any details on how this was measured, how many years of education or even what the highest education level was achieved in their different cohorts.

8) For figure 1, the overlap colour of green (between the red and blue AD groups) is too similar to the green used, one of these colours should be changed to make it easier for the reader to understand the figure.

Response to Review

We thank the reviewers for their careful review of the manuscript and their constructive comments. They have helped us to greatly improve the manuscript. A detailed response to each comment is included below, with the reviewer's comment in bold and our response immediately below.

Reviewer #1 (Remarks to the Author):

Felski et al. show that the proportion of activated microglia (PAM), quantified by morphological analysis in patient brains of the ROS/MAP study, is more discriminative of Alzheimer's disease (AD) status than other tested parameters. PAM improves logistic regression model performance compared to APOE ϵ 4 and is associated with histological hallmarks of AD e.g. total A β load and others. Moreover, the authors show that an increase in PAM parallels an increase in AD related pathological features as well as cognitive decline and is not associated with other age related pathologies. The authors claim their work confirms the higher importance of microglia activation over microglia proliferation in AD pathogenesis and a synergistic role of PAM and A β load via effects of PHF Tau to promote AD pathology. In follow-up high throughput analyses the authors show increased PAM to be associated with increased A β peptide, CIQA, DOSK2 and decreased VGF via proteomics, yet found no significant associations between PAM and transcriptome changes measured via RNA Sequencing. Using GWAS, the authors claim association of IT PAM with rs183093970 and show association of MF PAM with rs2997325T and claim it could be a new biomarker. LINC01361 was not affected by rs2997325T. The authors could show an association between PAM-linked genes (based on GWAS) and immunological signatures as well as known drug targets, yet could not confirm association between rs2997325T and microglia activation using TSPO PET. Finally, the authors claim that a genetic predisposition to activated microglia is a cause of AD pathogenesis and is reflected by their polygenic scoring-based methods.

The study is of importance to the microglia research community. It represents one of the largest studies analyzing microglia activation state in humans in the context of Alzheimer's disease and ageing so far. The work will influence thinking in the field, as it seems to confirm, yet with only one model, that both microglia activation and Amyloid β accumulation contribute to Tau pathology and eventually to cognitive decline.

Response: we appreciate the reviewer's enthusiasm for our work.

Major critiques

- **unclear description how the microglia were visualized and how many samples were analyzed**

Response: Full details of our method for microglial quantification can be found in the supplementary methods (page 3, paragraph 3). Sample sizes for microglial counts from each region are detailed in Table 1 with accompanying summaries of demographic statistics.

- **unclear reasons of choice of the particular way to detect microglia activation in brain samples**

Response: We thank the reviewer for an opportunity to clarify this important point. A key strength of our study is that we use the directly observed morphological characteristics of microglia to ascertain their activation state, as opposed to indirectly quantifying a proximal marker of microglial activation, such as proteomic and transcriptomic signatures (e.g. gene module expression in a tissue-based profile). Currently, there is no single marker that specifically marks “activated” human microglia, so alternative options are limited. To investigate this question, we examined the relationship between tissue-level transcriptomic measures of microglia in the section of our manuscript titled “Relationship of microglial activation to the cortical transcriptome and protein measures” (page 11, paragraph 3). These associations and their implications for AD pathology and aging are further explored in another publication by our group, which is cited in the manuscript on Line 318 (citation 34: Patrick et al., 2017, BioRxiv).

We acknowledge the caveats of measuring microglia based on morphological characteristics, which we discuss in response to another comment from the same reviewer, and we understand the limitations of our methodology. However, it is worth noting that, so far, microglial morphology captures the most robust association with aging-related neuropathologies of all microglial features. It is also considered to be the standard for a measure of microglial activation by neuropathologists. The state of the field is nicely summarized in a recent review on the topic, published May 18, 2018 (Neurobiology of Stress) by Bisht, Sharma, and Tremblay:

“Although not all research is based solely on microglia morphology to establish inflammation in the brain, the link between the two is so strong that only observing specific microglia morphology can evoke the consideration of pathology, which is probably true in most instances. As such, changes of microglia morphology can be seen as an effective way to assess their immune state, due to the specific morphological changes these cells go through when activated”

In order to make our reasons for our choice of microglial activation phenotype clearer in the manuscript, we have added the following sentence to our introduction, in paragraph 2:

“This morphological assessment of microglial activation stage represents a clear and robust measurement of neuroinflammation that cannot be captured by a surrogate marker.”

• no histological evidence that activated microglia and A β synergistically lead to cognitive decline via PHF Tau

Response: There is substantial literature showing the co-localization of neuritic amyloid plaques (which contain both amyloid A β peptide deposition and neurofibrillary tangles) and activated microglia using histology in animal models and in human Alzheimer’s brain tissue (Dickson et al., 1988, *Am. J. Pathol.*; Leibmann et al., 2016, *Cell Rep.*). It is not clear what type of evidence the reviewer would like to see in addressing his or her comment: by itself, histology cannot be used to demonstrate synergy between activated microglia and A β deposition in tau deposition and subsequent cognitive decline. Such an analysis would require longitudinal *in vivo* measures of all four variables, which remains difficult to do given current tau and microglial PET ligands. Short of this, measures derived from histological studies

can be used – as we have done – in statistical modeling to explore which model best explains the histological observations. Specifically, we have relied on statistical inference and standard causal mediation modeling algorithms to show that a synergistic interaction of amyloid deposition and microglial activation is the best fit of causal events leading to tau accumulation and cognitive decline in our data. Reviewer #2 raises a similar point regarding the inference of directionality in our causal model and we refer reviewer #1 to our response on that point. Proving that our model is correct is an important endeavor to be pursued, but it is outside the scope of the current manuscript and would require substantial resources to assemble the in vivo data described above in a reasonable sample size.

- **no mentioning of sample number for all high throughput analyses**

Response: We apologize for not delineating the sample sizes used in different analyses more clearly. Sample sizes for all analyses have now been reported in the manuscript in the following locations of the main manuscript:

Line 91: “Genotype array data for 2,067 ROS/MAP subjects was imputed using the Michigan Imputation Server (Haplotype Reference Consortium reference v1.1)”

Lines 104-106: “A total of 18 disease- and age-related neuropathologies were measured brain-wide ($n=985$), as previously published.¹⁶ A subset of up to 225 brain samples were also evaluated for the presence of microglia at three stages of activation in four regions...”

Lines 110-112: “longitudinal cognitive performance data available at the time of study. Scans of in vivo [¹¹C]-PBR28 binding were assessed in 27 subjects ($n_{CN}=13$, $n_{MCI}=7$, $n_{AD}=7$) as part of the IMAS study.”

Line 217: “...modules of co-expressed genes ($n=478$ participants) and 67 proteins of interest ($n=807$).”

Lines 218-219: “Proteomic analyses found increases of A β peptide to be associated with increased MF ($p=1.0\times 10^{-6}$, $n=187$) and IT PAM ($p=2.9\times 10^{-5}$, $n=184$), providing an independent...”

For polygenic score analyses, sample sizes of each analyzed GWAS are listed in Supplementary Table 7.

- **masking of microglia-specific transcriptome differences associated to PAM by application of bulk RNA Sequencing**

Response: This is a caveat that we acknowledge. We have cautiously interpreted our analyses of tissue-level RNA-sequencing data in this respect. Examining this question in data from purified microglia is an important endeavor that we have initiated (Olah et al., 2018, *Nat. Commun.*) by establishing an experimental pipeline to purify microglia from human autopsy tissue. However, since the pipeline requires fresh autopsy tissue, data accrual is slow, and we do not yet have a large enough dataset to meaningfully interrogate purified microglial RNA data in relation to AD-related traits. For now, we focus on the available tissue-level data which yields reasonably-powered analyses and clearly discuss the limitations of these analyses. We note that, while certain features of the microglia transcriptome are likely to be obscured in tissue-level data, we clearly recover microglia transcripts and microglia-driven co-expression patterns in the tissue-level data (Patrick et al 2018, *BioRxiv*).

- **differences found for VGF, CIQA and DOCK2 were not followed up**

Response: We thank the reviewer for their comment. Given the weakness of statistical effect of PAM on levels of VGF, C1QA, and DOCK2, we have not pursued further investigation of these targets in this study. Reviewer #2 had a similar comment, so we have moved this result to the Supplementary Material since the manuscript contains a large number of results and the main narrative is centered on the modeling of the histological data and genetic analyses of these data.

- **no differences in LINC01361 expression in the presence of rs2997325T**

Response: While we were not able to detect LINC01361 in our bulk brain RNAseq data (as indicated on line 249), we have described in the manuscript (lines 248-249; Table 2; Supplementary Table 5) that rs2997325 does determine the expression of LINC01361 in multiple peripheral tissues. This is part of the sum of evidence used to prioritize putative target genes in our GWAS-identified locus defined by rs2997325. These results help us to prioritize LINC01361 for further investigation by other investigators or ourselves in purified cell populations as part of future studies, and thus we decided to include the available data regarding the possible relation of this SNP and the expression of a nearby gene.

- **no confirmation of validity of rs183093970 and rs2997325T by TSPO PET**

Response: We were unable to test for an effect of rs18093970 on TSPO PET because of its low minor allele frequency (see lines 268-270): there were no subjects with the minor allele in the available TSPO dataset. However, a major strength of our manuscript is that we were indeed able to validate the effect of the common rs2997325T allele on TSPO PET in the same direction as would be expected based on our GWAS (see lines 270-273 and Figure 4i).

- **study seems overloaded by the high throughput data mentioned in a descriptive fashion**

Response: We are not entirely clear on what is meant by this comment; however, to avoid drawing focus on parts of the manuscript that the reviewer may deem descriptive we have shortened several sections and moved some results, such as the proteomic results, to the supplement. Our aim was to present a comprehensive evaluation of all available data to facilitate future work by members of the community. In response to this comment, we have attempted to streamline the manuscript's narrative.

- **the last part of the results is too descriptive**

Response: As mentioned above, we have made changes to this part of the results section of the manuscript to address the reviewer's concern and streamline the narrative of the manuscript.

- **putative bias by lack of information about pathological data of tested patients should be mentioned in the results part and ruled out by respectively reanalyzing the data**

Response: We unfortunately are not sure what the reviewer means by this comment. We have delineated the number of subjects used in the different analyses in response to a previous comment by this reviewer. Only subjects with complete phenotypic data (n=225) were used in our statistical modeling. No subjects with missing pathologic data were used.

Minor critiques

- **the number of samples analyzed are not mentioned in the figure legends**

Response: We have added sample sizes to all figure legends in the revised manuscript.

- **figure 4 g and i are not explained**

Response: Both Figure 4G and I are now explained in detail in the legend of Figure 4; However, we thank the reviewer for pointing out that they are not cited within the manuscript text. We have rectified this, and added citations for Figure 4g and 4i in the following locations:

Page 13, paragraph 2: “For MF PAM, a different locus on chromosome 1 reached genome-wide significance (top SNP: rs2997325^T $p=1.88 \times 10^{-8}$, $\beta=0.039$, S.E.=0.0066; Figure 4d,e,g).”

Page 14, paragraph 2: “In this slightly younger sample ($\mu_{\text{age}}=71.2$ years), we found that rs2997325^T was significantly associated with an increase in [¹¹C]-PBR28 binding in the entorhinal cortex *in vivo* ($p=0.02$, $r^2=17.1$; Figure 4i), consistent with our finding that the same allele increased MF PAM (Figure 4h).”

- **the results part is hard to read as it is too condensed in particular the first part of the results**

Response: In accordance with the reviewer’s previous comments, we have made changes to the language of some parts of the manuscript, which are highlighted in the revision, to streamline the narrative and avoid dense or overly descriptive segments.

- **discussion about putative bias through the applied quantification method for microglia activation and previous studies using this method including TSPO-PET is missing**

Response: We thank the reviewer for their comment. We have now included discussion of potential biases of our microglia quantification methods in the Discussion section of our manuscript:

Page 17, paragraph 2 – page 18, paragraph 1: “Finally, we acknowledge that our chosen methodologies for quantifying microglia have intrinsic biases and limitations. For example, the manual identification of stained microglia carries a degree of subjectivity and, while [¹¹C]-PBR28 is a well-validated biomarker of

microglial activation in humans, experimental confounds and the potential for non-specific binding are important to consider in any molecular imaging experiment.”

• outlook on how rs183093970 and rs2997325T could be further confirmed as new biomarkers is missing

Response: We thank the reviewer for this suggestion. We have added to the Discussion of our manuscript suggestions for further confirmation of these loci:

Page 17, paragraph 1: “Further investigation of cortical PAM genetic architecture is warranted to extend our initial set of observations; replication of associations with both histologic measures of microglial activation and TSPO imaging in populations at risk of AD are needed to assess the utility of these measures as biomarkers that may inform the diagnosis and/or management of aging-related cognitive decline. In addition, *in vitro* knock-down and DNA editing experiments, combined with amyloid and tau-sensitive assays, will be needed to explore the role of these variants in biological events related to AD.”

In conclusion, the authors show the association between the PAM score and pathological features of AD as well as cognitive decline. The statistical approaches and models used seem valid and appropriate. Covariates such as age at death, postmortem interval, APOEε4 status, and principal components are included in the described models.

Yet, it is necessary to rule out a putative bias caused by lack of information about patient pathological data that the authors mention in the discussion. The authors need to confirm their results by applying a further method to visualize microglia activation in order to rule out the respective bias of their method. In addition, the results of the proteomics, RNA- Sequencing and GWAS analyses should be followed up and confirmed further. The last part of the results can be left out to simplify the manuscript.

Response: We appreciate the reviewer’s comments. As noted in the response to a related comment above, we do not understand which “pathologic data” is missing since we only use subjects who have no missing pathologic measures in our analyses. Thus, there is no bias from missing data in our analyses. Also, importantly, the persons doing the cell counting are blinded to all clinical and pathologic data.

In terms of the comment to confirm the observation with an alternative measure of microglial activation, we respectfully disagree with the reviewer as the measure that we have used is an accepted standard that has been used by neuropathologists for decades (Hopperton et al., 2008, *Mol. Psych.*). There is, to our knowledge, no other histological measure with comparable properties as there is no other generally accepted markers that is specific for microglial activation in humans or is considered a proxy for these morphologic measures. Capturing data with such a marker would simply reflect the response of a single gene to the aging brain and require extensive validation efforts to assess its relevance to the gold standard: the morphology-based trait that we have used. Ultimately single cell level data will offer an evaluation of the type of that the reviewer suggests. While we have initiated such efforts, sample accrual is very slow and will not yield the necessary sample sizes to address this question for several years, making it impractical to be included in this manuscript.

Overall, we have used the gold standard, well-validated and widely-used measure of microglial activation in human neuropathology studies to drive our analyses, and we have provided evidence of replication for our genome-significant result using a related but completely different measure (TSPO imaging), which suggests that our results are robust. Given the nature of the measure that we have used and the lack of a clear, robust alternative measure to use in the type of validation effort that the reviewer suggests, we feel that no technical validation of our result is warranted as part of this manuscript. Future work with some experimental markers that may offer a complementary perspective on the role of microglia in AD is of interest but is beyond the scope of this manuscript.

The manuscript should be considered for publication after including the following experiments to support the claims of the authors:

- **reanalysis of the PAM associations excluding patient samples with incomplete information on pathological data**

Response: We are unsure what the reviewer is suggesting. We analyzed all outcomes in the sample of subjects for which full data were available. For the 225 subjects with PAM score data, postmortem pathological data and cognitive decline data were available for all.

- **reanalysis of PAM scores using a further method to analyze microglia activation e.g. Cd68, TMEM119 IHC**

Response: We have addressed this point in the response above. We note that CD68 and TMEM119 are general markers for all microglia, so we would not be able to use them to calculate the PAM.

While we appreciate that additional associations of molecular biomarkers with PAM could strengthen our findings, a follow-up quantification of cell surface markers in the same subjects in which PAM was quantified would be an endeavor beyond the scope of our current study. In order to assess the effects of PAM on proteins and mRNA transcripts that may canonically be involved in microglial activation, we have performed the proteomic and transcriptomic analyses that yielded nominal associations with VGF, C1QA, and DOCK2, as well as inflammatory gene modules. However, in total, the evidence largely pointed toward a lack of association of PAM with our proteomic and transcriptomic measures (final sentence of paragraph 1, page 12). This is likely due to the fact that proteins and RNA were measured in bulk tissue and – as discussed previously – the molecular profiles of active microglia are notoriously unstable and region-specific.

- **quantification of protein levels of VGF, C1QA and DOCK2 in patient samples to confirm association with PAM**

Response: We have included the results of our analyses with available proteomic measures such as the ones listed above to offer the community a comprehensive evaluation of all available data. However, as noted above, these proteomic results are not a major component of our narrative and have been shifted to the supplement to be available to the community. While replication of these results is of interest, the

necessary data are not currently available and would require substantial time and resources to be gathered. Since this is not a major component of the manuscript, replication is not necessary as part of this manuscript as we have focused our efforts on replicating the genetic result which is at the center of this manuscript.

- **if possible TSPO PET with more patients to confirm validity of rs2997325T and rs183093970 as biomarkers**

Response: Unfortunately, expanding the sample size for this analysis is difficult, as few TSPO PET imaging samples with full genomic data exist and such a PET scan costs in excess of \$5000 for each individual. Especially challenging is finding extension datasets within a similar age range and with similar clinical characteristics (i.e. healthy, mild cognitive impairment, and Alzheimer's disease). For future work, we are planning collaborations to image larger cohorts of individuals using TSPO radioligands; however, for the present study we are limited to the IMAS group, which we feel is sufficient for providing initial confirmation of the effect of rs2997325T on microglial activation.

- **in situ hybridization-based quantification of LINC01361 in patient brain samples**

Response: The genetic evidence supporting the role of LINC01361 in the effect of rs2997325T is modest at this point. We include it in our manuscript to be thorough and suggest avenues of future investigation. However, additional genetic studies are needed to support its role before we can justify performing in situ hybridization studies. Further, such staining would be difficult to interpret in the absence of a clear genetic link between LINC01361 and rs2997325T in the target tissue. We could remove this section since it has little bearing to the narrative of the paper and will defer to the editor on this point.

Reviewer #2 (Remarks to the Author):

This is an interesting paper that uses several analyses to investigate PAM (proportion of activated microglia) to neuropathology, genetics, and AD status.

Main comments:

1. I found the methods a little complicated and think that a flow diagram would be really helpful.

Response: We appreciate the reviewer's suggestion and agree that this would be helpful to many readers. We have added a new figure to the main manuscript accordingly. Page 6, paragraph 3:

"Figure 1 illustrates the progression of analyses performed in our study"

2. Why was a GWAS threshold of 2.5×10^{-8} used rather than a more liberal (but more widely applied) 5×10^{-8} threshold?

Response: We thank the reviewer for their attention to detail. This threshold was chosen as a correction on the widely applied 5×10^{-8} threshold for multiple comparisons (we performed two separate GWAS). We acknowledge that, due to the weak-moderate correlation of our outcomes for the two GWAS, the analyses are not entirely independent and thus the correction is conservative. Therefore, we established our suggestive criteria for GWAS results at a more liberal $p=1 \times 10^{-5}$ and performed our gene mapping based on this threshold. However, we do feel that the currently applied threshold is appropriate for concluding genome-wide significance given our study design, and, in any case, the results would be unaltered had we used 5×10^{-8} instead (i.e. no loci exceeding 5.0×10^{-8} do not exceed 2.5×10^{-8}).

3. Line 178: I don't think it's meaningful to describe an e.g., 18% improvement in model performance when considering AUCs. A better metric would be sensitivity, specificity, PPV and NPV.

Response: We appreciate the reviewer's feedback regarding reporting of discriminatory model performance differences. We feel that our choice of reporting change in AUC (Δ AUC, as a percentile) is informative, valid, and, importantly, familiar to the wide readership of Nature Communications. There is literature in support of this choice: Pencina and Delmer (2013, *Stat. Med.*) have recently evaluated the utility and validity of Δ AUC specifically in the context of "risk assessment algorithms introduced by the addition of new phenotypic or genetic markers", testing three metrics of discriminatory model improvement in the Framingham Heart Study (a study which closely resembles ROS/MAP in design). They conclude that while the information provided by Δ AUC is incomplete, it is useful and widely accepted. To clarify the metric in our paper, we have added the following to our Methods section under "Statistical analysis" (page 7, paragraph 1):

"Differences in model performance are reported as change in C-index, expressed as a percent, which corresponds to area under the receiver operating characteristics curve (Δ AUC)."

We agree with the reviewer that providing additional metrics of model performance and improvement would be beneficial to the interested specialist. Therefore, we have expanded Supplementary Table 1 to include sensitivity and specificity, as well as positive and negative predictive values for every model tested (optimal threshold determined by Youden's J statistic).

4. Line 182: for the comparison with APOE, what was the point estimate for e4e4 carriers? I appreciate that there won't be too many in the study but it would be interesting to report this. Also, was e4 status associated with pathology?

Response: This is an interesting question. Unfortunately, in the sample subset for which microglial count data are available, only two subjects are homozygous for *APOE* e4. As a result, we do not feel that reporting a point estimate of group effect would be meaningful. For the reviewer's own interest, these two e4 homozygous individuals have levels of PAM that are below the 30th percentile for all four regions tested, but, again, we caution against interpretation.

As for effects of *APOE* e4 status on pathology, there were strong effects (this can be seen on the ROC curves in original Figure 1B (now Figure 2C), where pathological AD is the binary outcome), which have been previously published in this cohort (Shulman et al., 2013, JAMA Neurology). Due to these well-established effects, we co-varied for *APOE* e4 status in all appropriate analyses to avoid confounding.

5. Line 198: What was the statistical model used for the cognitive decline analysis? On line 201 I find it strange that there is 'no sig effects of subcortical PAM on cognitive decline' when PAM measurement was taken after the cognitive data.

Response: For our analyses of cognitive decline, we first generated random slopes of longitudinal cognitive change using linear mixed models, as previously published. We then tested these slopes as outcomes of robust linear regression with our PAM measures as predictors. In this sense, we statistically were testing for effects of PAM on cognitive decline; however, as the reviewer points out, the phrasing suggests a temporal paradox. We have amended line 198 to be more accurate:

Page 11, paragraph 1: "Similar to our neuropathological findings, there were no significant associations of subcortical PAM with cognitive decline."

6. Paragraph from line 202: I think that the conclusions for this section are too strong. Given that PAM and pathology were measured at death, I don't think the authors can justify a PAM -> pathology -> cog decline pathway. Given the nature of the data collection, I don't think directionality can be assessed here.

Response: We thank the reviewer for their thoughts on this issue. We agree with the reviewer that formal causation cannot be demonstrated both because the cognitive data is collected prior to the pathology data and because the pathology data are cross-sectional. Validating the putative sequence of events that we propose is very important but will require substantial resources for longitudinal cognitive assessments coupled with longitudinal PET studies of microglia activation as well as amyloid and tau. Such a validation is not practical for this manuscript. However, our supported sequence of events is critical for the design of such a study: it generates a clear, testable hypothesis.

Where we respectfully disagree with the reviewer is on the value of the results of our analyses. Causal mediation modeling is a well-developed field of investigation with robust methods that produce rigorous results for generating hypotheses such as the one that we propose. Importantly, such modeling of the type of data that we have cannot provide formal evidence of causation, but it does produce reproducible, informative results. The causal mediation modeling that we performed in our data carries important caveats that we acknowledge in our manuscript. We have enhanced the discussion of these limitations and moderated the language relating to these results to present the results fairly. With these

changes, we feel that this analysis should remain as it provides important information that will contribute to the design of longitudinal *in vivo* studies.

To address the reviewer’s comments regarding the nature of data collection in this study and its impact on the inference of effect directionality, more specifically, we do appreciate that PAM and pathology are measured postmortem, while cognitive performance trajectories are measured over time before death. However, there are several reasons why we believe that cognitive decline, PAM, and pathology may be considered within the same mediation model: Cognitive decline is measured leading to death. Therefore, the last measurement of cognitive performance is within one year of autopsy for all subjects - postmortem interval is also very short for participants in our study subset (median hours from death to tissue preservation = 6.4, sd=7.8). As documented in many other manuscripts based on these pathologic and cognitive data, the pathological state of the brain at autopsy is highly reflective of the cognitive state of the subject in the months prior to death (Bennett et al, 2018, *J. Alz. Dis.*), and thus we feel justified in attempting to make inference on this pathway with these data, albeit with acknowledged caveats.

In order to demonstrate the validity of our claims, we performed additional analyses of PAM effects on cognition using only the cross-sectional cognitive score measured at the last visit prior to death as our outcome. In these linear regression models, which co-varied for age at assessment, *APOE* ϵ 4 status, sex, and years of education, we find that cognitive status at last assessment is even more strongly related to PAM than to slope of cognitive decline:

PAM region	Outcome	PAM effect term			Adjusted model r^2
		Beta	SE	p	
midfrontal	global cognition at last visit	-2.99	0.95	0.0018	0.1
	global cognitive decline slope	-0.28	0.094	0.0036	0.065
inferior temporal	global cognition at last visit	-4.31	1.04	0.0000489	0.13
	global cognitive decline slope	-0.34	0.1	0.0014	0.078
ventral medial caudate	global cognition at last visit	-1.88	1.45	0.19	0.047
	global cognitive decline slope	-0.08	0.14	0.59	0.018
posterior putamen	global cognition at last visit	-0.77	1.22	0.53	0.058
	global cognitive decline slope	-0.027	0.12	0.82	0.029

Thus, overall, we feel that our modeling provides an important contribution to the manuscript using robust, well-accepted methods for causal mediation modeling. By clearly discussing the caveats of our results, we feel that we have appropriately outlined the limitations of the data and methods; further, we have moderated our interpretation of these results to avoid giving the impression that the sequence of events relating microglial activation, amyloid, tau and cognitive decline is settled.

7. Line 222: I don't find the nominal (non-significant) reporting to be helpful unless there is external replication.

Response: We agree that in the absence of external replication these results should be considered as exploratory. Considering that the study we cite (Olah 2018, *Nat. Comm.*) identified these same genes

(C1QA and DOCK2) in a separate sample of postmortem samples using a different methodology (RNAseq data generated from purified microglia), our finding represents a form of external validation, albeit independently non-significant. Nonetheless, we appreciate that the reporting of these results in the main manuscript may not be helpful, so we have moved these results to Supplementary Material, page 6:

“In addition to significant effects of PAM on levels of VGF, nominal positive associations ($p < 0.05$) of subcortical PAM were observed for two genes in prefrontal cortex (CIQA and DOCK2) which were recently identified as strongly upregulated in the transcriptome of purified aged human microglia from the same cohort.³⁴”

8. Line 238: I am not convinced by the rare variant finding. A MAF <1.5% in such a small cohort does not seem appropriate as outliers will drive associations at this threshold. I think that a more strict MAF and MAC should be applied here.

Response: We agree that the uncommon variant finding should be viewed cautiously, and we have made a point to emphasize the caution with which we report it. We chose to keep the result in our study in the interest of discovery, since follow-up studies will now be able to test it. It is worth noting that the results are significant at a genome-wide threshold and driven by seven heterozygous individuals, rather than only one or two outliers. Thus, there is some robustness to this result; we note in the manuscript that all seven individuals possess PAM scores in the top 95 percentile.

9. Line 250: What was the genetic correlation between the two PAM GWAS?

Response: Genetic correlation using LD score regression is not possible given our low sample sizes as heritability estimates and thus LD regression coefficients are not interpretable. However, to address the reviewer’s question, we performed the same polygenic scoring method applied to other GWAS traits in

our study using PRSice to test the strength of association between inferior temporal and midfrontal PAM genetics. The results to the left:

Figure R1: This plot shows that, for each microglial activation GWAS, the variance explained (pseudo r^2) in the other GWAS increases to nearly 100% (upper threshold 1×10^{-4}). IT = inferior temporal cortex; MF = midfrontal cortex.

While the genetic association between the two GWAS is strong, it is not unexpected given the phenotypic correlation and the fact that the phenotypes were measured in the same

group of subjects. It is interesting that there are some differences in polygenic associations of IT and MF microglial activation with several published traits despite this apparent similarity in their genetic underpinnings. This is due to the fact that at lower p-value thresholds (e.g. 1×10^{-5}), the correlation (r) between MF and IT microglial activation is less than 0.5 ($r^2 < 0.25$). Supplementary Table 4 includes the summary statistics for all significant and suggestive loci for both GWAS, so that the reader can quickly assess the effects of all SNPs in either GWAS.

10. Line 264: are any of these drugs relevant?

Response: The reviewer raises an important question. The answer is yes, several of the drugs mentioned in the manuscript and detailed in Supplementary Table 8 are relevant to immunological and neurological traits and diseases. Examples include Acitretin, which targets the *RXRG* gene product and is used to treat psoriasis, and Eletripan, which targets HT2RB and is used in the management of migraine. We have added a line to page 14, paragraph 1 to bring this to the reader's attention:

“Of these, some, such as Acitretin and Eletripan, are relevant to immunological and neurological illness; they are used in the treatment of psoriasis and migraine, respectively.”

11. Line 272: how many statistical tests were performed here?

Response: These analyses were performed in two steps – first as a multivariate test for all ROIs (which were strongly correlated) and then as post-hoc tests to identify the strongest individual effect. Thus, the p-value that we report for left entorhinal cortex is uncorrected, the omnibus multivariate test ($p=0.024$) indicates significant associations within the selected regions of interest, chosen for their susceptibility to pathologies linked to microglial activation: bilateral entorhinal cortex, hippocampus, and medial temporal lobe. The regression method is a reverse regression (SCOPA: Software for COrelated Phenotype Analysis) described and implemented by Mägi et al., 2017 (*BMC Bioinformatics*), which models allelic dosage as a linear function of correlated phenotypes to account for collinearity after the removal of technical and biological confounders (as listed in the manuscript).

We agree with the reviewer that these details are of interest and should be included in the manuscript. We have therefore added the following description to the supplementary methods section on “[11C]-PBR28 PET image acquisition and analysis”; Supplementary page 4, paragraph 3:

“For these analyses, we used three regions of interest (ROIs) for each hemisphere separately, resulting in six brain regions. First, we performed linear regression analyses using the six ROI values separately and identified an association in the left entorhinal cortex (uncorrected p -value < 0.05). As six ROI values were strongly correlated and the sample size is moderate for genetic analysis of this type ($N=27$), we carried out a multivariate analysis by performing genetic association analysis using six phenotypes simultaneously to minimize the number of test performed and increase the statistical power. We used the SCOPA (Software for Correlated Phenotype Analysis) program¹⁶ to perform an aggregation of genetic analyses for our multiple correlated phenotypes. The multivariate analysis of six correlated phenotypes yielded a significant association of rs2997325 with microglial activation (corrected p -value = 0.024).”

12. Paragraph from Line 275: the predicted r^2 values here are extremely small $<0.1\%$. What sort of effect sizes are observed if you simulate under the null?

Response: We acknowledge that these r^2 values are small, demonstrating the statistical difficulties in modeling genetic overlap in studies of relatively small sample size. As such the analysis is intended to reveal relationships between genetic etiologies of common disorders and PAM, rather than to develop strong predictors of risk in the general population (i.e. our focus is on the hypothesis test rather than the effect estimation).

Simulating under the null hypothesis of no relationship between PAM and trait genetics (i.e. permuting the p-values of our PAM GWAS and re-running the scoring) reveals expectedly smaller effect sizes than we observed in our polygenic testing of unpermuted data. I have demonstrated this below by performing 100 permutations at eight different score p-value thresholds (the number of SNPs is shown on each plot facet and increases as the p-value threshold increases), providing a series of null distributions for pseudo- r^2 . As can be seen, the unpermuted values of pseudo- r^2 (red lines) are in fact lower than random permutations at the lowest p-value thresholds. However, between $p < 0.0001$ and $p < 0.001$ ($n_{SNPs}=468$ and $n_{SNPs}=4893$, respectively), this pattern changes strikingly, indicating that as more variants in the unpermuted data are included in the score, the effect becomes much larger than expected under the simulated null:

Response Figure 2. This plot shows simulated null distributions for GRS associations for the midfrontal cortex and Alzheimer’s disease. The numbers labeling each facet indicate the number of SNPs included in each score (corresponding to p-value thresholds of $1e-6$, $1e-5$, $1e-4$, $1e-3$, 0.01 , 0.05 , clockwise from top left). Unpermuted data effect shown as vertical red line.

The calculation yielding pseudo- R^2 for the risk score model is from the ‘grs.summary’ function in Toby Johnson’s ‘gtx’ package:

$$R^2_{rs} < 1 - \exp(-X^2_{rs}/n)$$

This calculation relies on both the magnitude of the test statistic (X^2_{rs}) and the sample size of the target GWAS (n) such that the higher the sample size, the less variance is explained. This method has been applied in large-scale GWAS and described in further detail by Dastani et al., 2012 (PLOS Genetics).

13. From line 292: as with previous comments, I wonder how the timing of the PAM measure (at post-mortem) will impact interpretation in causal modelling. I think this is complicated and needs a caveat.

Response: We thank the reviewer for their comment. We understand that our mediation analyses (scrutinized above), which model phenotypic data alone, carry with them certain limitations with respect to causal inference. However, our polygenic analyses, which operate on principles of Mendelian randomization, are not subject to the same temporal caveats, since genetics are set at birth. At this point in the manuscript, the phenotypic relationships are determined based solely on genetic causes. Thus, we feel that our conclusions (also cautiously presented) are justified.

14. The ROS data are from "across the United States" but site information does not seem to have been included in the modelling. Is the random intercept term for site non-zero in a model with PAM as the outcome?

Response: All subjects with PAM measures were from the memory and aging project (MAP) which only recruits subjects in the Chicago metropolitan area; thus site is not a confounder in these data. We have added a sentence to the methods to clarify this. The reason for including both studies in the methodological description is that data from both are pooled, and published data are available, as indicated, via the Rush Alzheimer's disease sharing hub under the name ROS/MAP.

Minor comments:

Line 54: typo "a molecularly defined subtypes"

Response: We have fixed the typo.

Line 66: I am not a fan of using the word 'unique'. I don't think it adds anything here.

Response: we have removed the word "unique".

Line 82: of "the" elderly?

Response: we have added the missing word.

Line 90: typo "Tanscriptomics"

Response: We have corrected this typo.

Line 106: why were these regions selected?

Response: The data used in our analyses were generated as part of a different study examining the relation of dietary factors to neuroinflammation. We repurposed the data to conduct the analyses presented in this manuscript, which have not been reported elsewhere. Thus, we did not select these brain regions for this project, but they are relevant to the study of cognitive decline in older individuals. We are currently initiating data collection on additional regions of interest. However, these data are not yet available.

Line 110: it would be good to know the list of 17 cognitive tests that were used. This info is not present in the supplement.

Response: We have included a new supplementary table (Supplementary Table 10) listing the specific cognitive tests used to evaluate each domain of cognitive performance in the ROS/MAP studies. We have also added the following text to our Supplementary Methods:

Supplementary page 4, paragraph 1: “The full list of tests is provided in Supplementary Table 10”

Line 125: typo co-varying

Response: We have corrected the typo.

Line 126: PMI - please spell out

Response: We have spelled out the acronym.

Line 129: please list the suite of analyses that were performed using FUMA. 'The full complement of state-of-the-art tools' is not clear.

Response: We have listed in detail each of the analyses that were conducted using FUMA in our supplement (Supplementary Methods page 5 paragraph 6 through page 6 paragraphs 1 and 2), though we understand that the limited information in the main manuscript may be unclear. We have therefore expanded this description as follows:

Page 7, paragraph 2: “Analyses included positional and eQTL-based gene mapping, assessment of Combined Annotation Dependent Depletion (CADD v1.3)²² and RegulomeDB scores,²³ tissue expression specificity using GTEx v7, comprehensive gene set enrichment analyses, and mapping of gene targets to the DrugBank database.²⁴”

Line 154: typo (S)pearman

Response: We have corrected the typo.

Line 158: please state the n's (or n range) for the much smaller studies

Response: We have included the n range for studies of comparable design according to the Hopperton et al. (2018, *Mol. Psych.*) systematic review cited on Line 158.

Line 160: Why were Welch t-tests used? The data don't look terribly Gaussian. I think Mann-Whitney tests might be more appropriate although this shouldn't change the results.

Response: Welch t-tests were used to avoid the assumption of equal variance, though, as the reviewer suggests, the Mann-Whitney test would be more appropriate given the distribution of the data. That said, and as the reviewer is no doubt familiar with, given our sample size, the Welch t-tests are quite robust to the assumption of normality, and our results are not different when using the non-parametric equivalent.

Line 193: PHF - please spell out

Response: We have spelled out the acronym.

Line 193: There are 18 pathologies in Figure 2a but 14 reported on line 191. They're explained more fully in the supplementary methods but having a sentence about what the measured neuropathologies were somewhere in the main text might be helpful.

Response: We thank the reviewer for their attention to detail. Line 191 is in fact a typo, a result of an earlier iteration of the analyses that did not include four newer pathologies. We have remedied this. To briefly describe the nature of the measured pathologies in our main methods section, we have added a line in page 6, paragraph 2:

“Pathologies included multiple validated measures of A β peptides, neuritic and diffuse plaques, hyperphosphorylated tau protein, neurofibrillary tangles, micro and macro cerebral infarcts, cerebral atherosclerosis, TDP43 proteinopathy, and hippocampal sclerosis, among others (see Supplementary Materials for details)”

Line 194: 'increase' suggests that changes in PAM have been observed. Maybe replace with 'higher'

Response: We appreciate the reviewer's suggestion and have made this substitution, which is indeed more accurate.

Line 247: 8.9 kb 3' - is there something missing here?

Response: This phrasing is intentional; 3' refers to the genomic direction, indicating the relative location of rs2997325 to *LINC01361*.

Line 294: Please add a ref to Figure 5 as this is the first mention of SCZ

Response: Figure 5 includes many new abbreviations, defines them, and refers to the supplementary material for further detail such as GWAS publication citations. We have chosen not to include the citations for all of these studies in the figure legend to avoid unnecessary clutter.

Line 313: It might be interesting to expand this a bit and relate it to the differences in cortical/subcortical pathology

Response: We agree with the reviewer that this statement can benefit from some expansion, so we have added the following to our Discussion, page 16, paragraph 1:

“Further work now needs to be conducted to sample a wider array of regions in individuals with a range of pathologic burden and those without pathology so that we can better understand the extent to which these regional differences may be related to the nature of the brain region itself or the extent to which that region is affected by amyloid or tau pathology in a given individual. Currently, the data are too sparse to be able to interpret our results beyond simply noting that PAM in subcortical regions do not appear to relate to the burden of AD pathology.”

Reviewer #3 (Remarks to the Author):

Neuropathological correlates and genetic architecture of microglial activation in elderly human brain

This paper has examined the microglial morphology in the elderly and compared the proportion of activated microglia found in the brain with markers of Alzheimer’s-related dementia (i.e. amyloid- β , tau and rate of cognitive decline). The authors also performed genome-wide analysis to identify a common variant (rs2997325) that impacts microglial activation. The authors of this study have primarily used mathematical modeling to investigate and interpret their findings. This is an interesting paper, but would benefit from additional figures and descriptions.

1) The microglia are divided into three activation stages; stage I (thin ramified processes), stage II (plump cytoplasm and thicker processes), and stage III (appearance of macrophages). This reviewer finds it very surprising that we never see an image from the histological analysis to demonstrate these three subgroups. There is no information on the staining protocol or the stain(s) used on the slices, the analysis protocol is completely missing (how many sections were analysed per individual or how many cells), the factors used to define the processes as thick or thin is absent (was this a program? By eye? Based on size i.e. nm etc?). We see no quantification of the histology results. This is all required for the paper as these findings are the basis on which the entire mathematical modeling is founded.

Response: The detailed methods on microglial quantification and visualization are found in supplementary methods page 3 paragraph 3. Histopathological images of each stage of microglial activation in our cohort can be found in previously published materials (e.g. Patrick et al., 2017, BioRxiv

(Figure 3); Bradshaw et al., 2013, Nature Neuroscience (Figure 3C)). We initially refrained from including them in this paper to avoid repetitious republication of these images. Nonetheless, we agree that it would be helpful to readers to have instant access to these images, so we have modified Figure 1 to include an additional panel (now Figure 1A) showing representative histopathological scans of each microglial activation stage in our sample.

2) The third stage of microglia activation is also troubling to this reviewer, i.e. the “appearance of macrophages”. As we have no idea what stain or antibodies are used to define macrophages in the brain, one can only imagine how this cell population was distinguished from the resident microglial cells. It is very difficult to distinguish microglia from macrophages, as they are both cells of the myeloid lineage they share many similar receptors and markers. How were the macrophages removed from the analysis to only focus on microglia? This is a critical point that needs to be addressed with suitable example pictures for the paper.

Response: The reviewer brings up an important and largely unresolved question in the field as there are currently no markers that uniquely discriminate human microglia from infiltrating macrophages, particularly when these cells are activated. We are beginning to address this question by generating single cell data from purified human microglia from the same midfrontal cortical tissue sample that is studied in the current manuscript (Olah et al., 2018, *BioRxiv*). This independent effort is helpful in showing that >98% of myeloid cells in the brain of aging individuals appear to be microglia; myeloid cells with gene expression features consistent with bone-marrow derived cells account for ~1% of the observed myeloid cells in this report. Thus, the vast majority of myeloid cells that we are counting in the current manuscript are likely to be microglia and not cells derived from infiltrating peripheral blood cells. This single cell analysis also reports the presence of 14 distinct clusters of microglia in the 11 individuals profiled to date. However, it remains unclear which of these 14 microglial subtypes are “activated” or may relate to morphologically-defined stage III microglia; the reality is more complex as we need to recognize the heterogeneity of these cells. Much work remains to be done and a novel toolkit deployed in a large number of samples is needed to understand how these features relate to one another.

Thus, macrophage, if they are present in our tissue section, are not distinguishable from microglia, and there is no marker that could address this question effectively as all existing markers have caveats that would preclude a definitive resolution to this question. However, macrophages, as noted above, are, at best, a very small fraction of the observed cells. Thus, they will not meaningfully influence the results of our analyses. In any case, if the macrophages are creating statistical noise in the analysis, this will tend to reduce the true effect of microglia, which would therefore be actually larger than the ones that we report. Even if we remove 2% of counted cells from the total of stage III microglia, assuming that all macrophages are activated, the results remain highly significant.

Nonetheless, to highlight this important caveat, we have altered our discussion as follows:

Page 18, paragraph 1: “Further, we recognize that infiltrating macrophages present in brain tissue samples may represent a source of error in our counts of largely indistinguishable microglia.

Reassuringly, recent single-cell analyses performed by our group have shown that these cells represent a negligible minority (~1%) of myeloid lineage cells present in brain tissue from similar subjects.”

3) There are a number of grammatical errors throughout the manuscript, and abbreviations are sometimes used without introducing them first. The sentence running from page 13 to 14 is incomplete.

Response: We thank the reviewer for their attention to detail and apologize for these errors. We have made the needed corrections indicated by highlight in the revised manuscript, including the incomplete sentence running from page 13-14.

4) Page 9 sentence 171, 172 states microglia activation rather than proliferation mediates neurodegeneration but the paper referenced is from 2002 and since then there have been other reports demonstrating the microglia proliferation does have a role in neurodegenerative disease development, e.g. Gomez-Nicola et al 2013 “Regulation of Microglial Proliferation during Chronic Neurodegeneration”. This should be updated in the text.

Response: We thank the reviewer for their comment. We have updated reference #26 as suggested.

5) It is not clear what cognitive tests were performed on participants to determine cognitive decline, and AD diagnosis. A table should be included.

Response: See responses above - we have included a new supplementary table (Supplementary Table 10) which contains details relating to neuropsychologic testing.

6) It would be beneficial to see the quantification of tau and amyloid on the different disease groups, before being used in the regression studies. This could also be included as a table.

Response: We have included another supplementary table (Supplementary Table 9) breaking down the distributions of total amyloid and paired helical filament tau in the AD and non-AD disease groups. We have also added the following text to our Supplementary Methods:

Supplementary page 3, paragraph 1: “Supplementary Table 9 describes the distributions of tau and amyloid neuropathologies in the pathologically confirmed AD and non-AD groups separately for both the entire available ROS/MAP cohort and the sample subset for which postmortem microglial density data were available.”

7) The authors use “educational attainment” as an effect in their modeling, yet they do not provide any details on how this was measured, how many years of education or even what the highest education level was achieved in their different cohorts.

Response: The educational attainment phenotype was an outcome of the genome-wide association study performed by Okbay et al. (2016, Nature; PMID 27225129), and it is described in detail in that study. In order to avoid duplication of published methods we have not re-printed the ascertainment methods of each GWAS that was used in the polygenic modeling section of our study since 29 studies are considered. We would refer the specialist to each particular manuscript for the fine details of their methods. However, given that educational attainment was a focus of our results and is mentioned in the discussion, we have added some detail from the aforementioned manuscript to our Supplementary Material to allow the interested reader to quickly access details on how this phenotype was measured in the Okbay et al., 2016 meta-analysis:

Supplementary page 6, paragraph 4: “Our results pinpoint educational attainment as an important phenotype related to the genetic architecture of microglial activation. The published study from which educational attainment GWAS statistics were extracted is Okbay et al., 2016.³⁴ The phenotype was calculated based on a standard in the 1997 International Standard Classification of Education (ISCED) of the United Nations Educational, Scientific and Cultural Organization. Due to the diversity of study settings from which meta-analyzed samples were collected, it was necessary for authors to map each major educational qualification that it is possible to attain in a specific country into one of seven harmonized ISCED categories. Their main outcome variable was then imputed as a years-of-education equivalent for each ISCED category.”

8) For figure 1, the overlap colour of green (between the red and blue AD groups) is too similar to the green used, one of these colours should be changed to make it easier for the reader to understand the figure.

Response: The coloring of the figure is based on translucency of the two colors used in plotting the densities of each distribution. We did not intend for a third ‘overlap’ color to be included, as the coloring is referring to two non-overlapping groups (AD vs. non-AD). To aid in discriminating between these two groups by increasing color contrast, we have modified the color scheme of Figure 1A (which is now Figure 2B).

REVIEWERS' COMMENTS:

Reviewer #1 (Remarks to the Author):

The authors met mostly all major critiques and stratified the manuscript. The overly descriptive parts are reduced and the main messages are clearer now.

The authors explained the time and cost intensive nature of my suggestions for additional experiments. Therefore, I can see why they are not within the scope of this work. However, the authors should discuss how i) reanalysis of PAM scores using a further method to analyze microglia activation and ii) TSPO PET with more patients to confirm validity of rs2997325T and rs183093970 as biomarkers would add robustness to the clinical applicability of the PAM score. In addition, a discussion of how single cell sequencing of microglia and the PAM score could be combined in future studies to validate the association of PAM with microglia activation states in different regions should be included.

Reviewer #2 (Remarks to the Author):

I thank the authors for their detailed responses and have no further suggestions.

Reviewer #3 (Remarks to the Author):

None

REVIEWERS' COMMENTS:

Reviewer #1 (Remarks to the Author):

The authors met mostly all major critiques and stratified the manuscript. The overly descriptive parts are reduced and the main messages are clearer now.

The authors explained the time and cost intensive nature of my suggestions for additional experiments. Therefore, I can see why they are not within the scope of this work. However, the authors should discuss how i) reanalysis of PAM scores using a further method to analyze microglia activation and ii) TSPO PET with more patients to confirm validity of rs2997325T and rs183093970 as biomarkers would add robustness to the clinical applicability of the PAM score. In addition, a discussion of how single cell sequencing of microglia and the PAM score could be combined in future studies to validate the association of PAM with microglia activation states in different regions should be included.

Response: We appreciate the reviewer's comment and agree that the points they raise should be discussed in the paper. We believe that reanalysis of PAM scores using additional methodologies and replication samples will be essential to solidify it as a robust and clinically useful biomarker. In response to an earlier comment, we included the following passage in our Discussion which we feel encompasses the first two points raised by this reviewer:

Page 14, paragraph 1: "Further investigation of cortical PAM genetic architecture is warranted to extend our initial set of observations; replication of associations with both histologic measures of microglial activation and TSPO imaging in populations at risk of AD are needed to assess the utility of these measures as biomarkers that may inform the diagnosis and/or management of aging-related cognitive decline."

Regarding this reviewer's comment on single cell sequencing, we are currently pursuing such experiments in purified microglia as part of a separate study, where we propose that newly identified human microglial subpopulations may map onto the PAM phenotype. As such, we have added the following passage to the Discussion:

Page 15, paragraph 1: "These same single-cell sequencing experiments will be important for identifying molecular subpopulations of microglia that may correspond to those activation states captured by the PAM phenotype."

Reviewer #2 (Remarks to the Author):

I thank the authors for their detailed responses and have no further suggestions.

Reviewer #3 (Remarks to the Author):

None